# Müller glial microRNAs are required for the maintenance of glial homeostasis and retinal architecture

Stefanie G. Wohl [1], Nikolas L. Jorstad[1], Edward M. Levine[2] & Thomas A. Reh[1]

To better understand the roles of microRNAs in glial function, we used a conditional deletion of Dicer1 (Dicer-CKO$_{MG}$) in retinal Müller glia (MG). Dicer1 deletion from the MG leads to an abnormal migration of the cells as early as 1 month after the deletion. By 6 months after Dicer1 deletion, the MG form large aggregations and severely disrupt normal retinal architecture and function. The most highly upregulated gene in the Dicer-CKO$_{MG}$ MG is the proteoglycan Brevican (Bcan) and overexpression of Bcan results in similar aggregations of the MG in wild-type retina. One potential microRNA that regulates Bcan is miR-9, and overexpression of miR-9 can partly rescue the effects of Dicer1 deletion on the MG phenotype. We also find that MG from retinitis pigmentosa patients display an increase in Brevican immunoreactivity at sites of MG aggregation, linking the retinal remodeling that occurs in chronic disease with microRNAs.

[1] Department of Biological Structure, University of Washington, Health Sciences Center, Box 357420 ,1959 Pacific Street NE, Seattle, WA 98195, USA. [2] Department of Ophthalmology and Visual Sciences, Vanderbilt University, Nashville, TN 37232, USA. Correspondence and requests for materials should be addressed to T.A.R. (email: tomreh@u.washington.edu)

Müller glia (MG) are the predominant type of retinal glial cell and along with maintaining tissue homeostasis and providing support and protection for neurons, they are required for retinal structural integrity[1, 2]. Several studies have shown that loss of mature MG in a range of species leads to disruptions in retinal structure[3–5]. Moreover, after loss of neurons, MG respond to the insult with changes in expression of cytoskeletal genes (e.g., GFAP) and a reduction in genes associated with their normal functions. In cases of chronic and progressive neuronal loss, hypertrophic MG migrate from their normal position in the inner nuclear layer (INL) and are associated with large-scale neuronal disorganization[6]. The MG migration and neuronal disorganization that results from chronic retinal damage, is a potential limit to current attempts to restore retinal function by transplantation, gene therapy, or prosthetic devices, but little is known about the factors responsible for the migratory behavior of MG.

microRNAs (miRNAs) are important regulators of gene expression in development[7–12], but also play roles in disease and degeneration[13, 14]. In the brain, predominantly for astrocytes, miRNAs have been reported to regulate injury responses[15–18] and are involved in cell degeneration and tumor genesis[15, 19–22]. A number of studies analyzed the effects of Dicer1 deletion in glial development[17, 23–27], but the role of miRNAs in mature glial function are not as well characterized.

To better understand the cellular processes regulated by miRNAs in glia, we carried out a MG-specific deletion of Dicer1 (Dicer-CKO$_{MG}$), an endoribonuclease required to produce mature miRNAs[28]. The result of the loss of Dicer1 in differentiated MG is a temporary increase in their number and their migration to ectopic positions in the retina. After 6 months, we observe a decline in MG number and the formation of MG aggregations in vivo, and a disruption of the retinal architecture in the Dicer-CKO$_{MG}$ mice. There is also significant impairment of visual acuity in the Dicer-CKO$_{MG}$ mice. The loss of Dicer1 in MG results in a decline of all the miRNAs that are normally highly expressed in these cells, and RNA-Seq shows that the gene with the greatest increase in the Dicer-CKO$_{MG}$ is Bcan (encoding Brevican), a chondroitin sulfate proteoglycan[29, 30]. Additional studies further support a role for Brevican and miR-9 in MG aggregation and migration. Together, our results show that miRNAs are required in MG for the maintenance of retinal structure and function.

## Results

**Dicer1 deletion in MG disrupts normal retinal architecture.** In order to analyze the role of miRNAs in MG, we used a MG-specific CreER line (Rlbp1-promoter driving an estrogen receptor-cre-recombinase) crossed to a stop$^{f/f}$-tdTomato reporter mouse (Fig. 1a). After injection of tamoxifen (Methods), tdTomato$^+$ cells were exclusively found in the INL, expressing the glial markers Sox9 and glutamine synthetase (GS) (Fig. 1c–f) as reported before[31, 32]. We did not find tdTomato expression in retinal neurons or astrocytes (Supplementary Fig. 1a–t), in accordance with previous studies[33]. By crossing the inducible Rlbp1CreER with mice with floxed alleles of Dicer1 (exon 23; the second RNase III domain)[28], Dicer1 deletion was effected with four consecutive daily injections of tamoxifen in young mice (P11–14, after retinogenesis is complete and MG have differentiated[34–36], Fig. 1a, b).

One month after the last tamoxifen injection, we found distinctive changes in retinal architecture in the Dicer-CKO$_{MG}$ mice. Although the overall structure of the retina was intact, the MG in the Dicer-CKO$_{MG}$ retinas migrated to abnormal locations in the outer plexiform layer, the outer nuclear layer (ONL), and

the inner plexiform layer (Fig. 1g–j, w; Supplementary Data 1). By contrast, Dicer1 heterozygous mice were not different from wild-type (wt) in retinal structure and glial cell numbers (Supplementary Fig. 2a–i).

In the more severely affected regions, there was a 45% increase in INL thickness in the Dicer-CKO$_{MG}$ retinas compared to the wt (Supplementary Fig. 3a–e). Cell counts revealed that the INL of Dicer-CKO$_{MG}$ mice had ~23% more tdTomato$^+$ MG than the wt and all tdTomato$^+$ cells were co-labeled for the MG markers Sox9 and glutamine synthetase (GS, Supplementary Fig. 3a–d, f). In many regions, in the Dicer-CKO$_{MG}$ retinas, the MG were not aligned in a single row, but distributed over the entire INL (Supplementary Fig. 3a–d, g).

Six months after Dicer1 deletion, more severe alterations in the retina were observed (Fig. 1k–v). MG were nearly all misplaced from their normal location, and often formed large aggregations (Fig. 1q–s); 38% of all MG were now located in the ONL (Fig. 1w; Supplementary Data 1). In addition, the retina was much thinner than normal and significantly disorganized. Also, by 6 months after the Dicer1 deletion in the MG, there were fewer MG than normal, in particular in the central retina (Fig. 1q–v, x, y; Supplementary Data 1). Interestingly, 96% of the tdTomato cells were still Sox2$^+$, but only 19% expressed GS (Supplementary Fig. 3h, i). The formation of aggregates was not as prominent in the peripheral retina, and the number of Sox2$^+$ MG were similar to the wt; however, there were clear abnormalities in MG position and many MG had very low levels of GS (Fig. 1t–v, y; Supplementary Data 1).

Dicer1 deletion in the MG also led to disruption of the outer limiting membrane (OLM), which is formed by MG (Fig. 2a–e). This was accompanied by an obvious reduction of ONL thickness and in the number of layers of Otx2$^+$ cells in the ONL in the center and periphery, 6 and 12 months after Dicer1 deletion (Figs. 2, 3a–c, Supplementary Data 1). To assess the timing of neuronal cell death in the mutants, we used Caspase3 labeling. We did not detect any Caspase3$^+$ cells 1 month after Dicer1 deletion. However, 6 and 12 months after Dicer1 deletion, we found Caspase3$^+$ cells in the INL and GCL, suggesting that loss of Dicer1 in MG indirectly causes neuronal death in later stages (Fig. 2f–l). Interestingly, 1 month after Dicer1 deletion, the ONL was significantly larger than in corresponding controls (Supplementary Fig. 3j, k). However, since we did not find an increased number of Otx2$^+$ cells, the extended ONL might be a result of a more stretched retina, which could be due to the loss of MG tensile strength and the resulting overall loss of structural integrity[4].

Since the retinal structure was disrupted in the Dicer-CKO$_{MG}$ retinas, we analyzed the visual acuity of the mice using OptoMotry. For the wild-type mice, we found visual acuity values of 0.4 cycles per degree (Fig. 3d; Supplementary Data 1) in accordance with previous reports[37]. In the Dicer-CKO$_{MG}$ mice, however, the visual acuity was significantly reduced as early as 3 weeks following Dicer1 deletion and further declined to 0.2 cycles per degree by 28 weeks. The reduced visual acuity at P28 is an interesting finding since we did not find any obvious phenotype at this time. However since full visual acuity is reached around P25[37], the loss of Dicer1 in the MG induced at P11–14 might have an impact on retinal maturity.

The changes we observed in the MG after Dicer1 deletion in young mice are relatively slow and initial experiments in which Dicer1 deletion was initiated in mice older than P21 failed to show a phenotype after 1 month. However, more extended analysis of mice in which Dicer1 was deleted in older animals (60 days old) showed a phenotype at 2 and 5 months after cre induction (Fig. 3e, i), similar to that observed in the P11–14 induced mice. Two months after Dicer1 deletion, the number of

tdTomato[+] MG was significantly increased and the INL was extended in the Dicer-CKO<sub>MG</sub> compared to the wt (Fig. 3f–h; Supplementary Data 1; Supplementary Fig. 3l–n). Five months after Dicer1 deletion, we found a significant disruption of the retinal architecture in both, central and peripheral retina (Fig. 3i;

Supplementary Fig. 3o). The phenotype was not as severe as in the mice that had Dicer1 deletion at P11–14, although there was a similar decline in the ONL and disruption of the OLM. The progression of retinal disorganization was slower when Dicer1 was deleted in adult mice than from a similar deletion in young

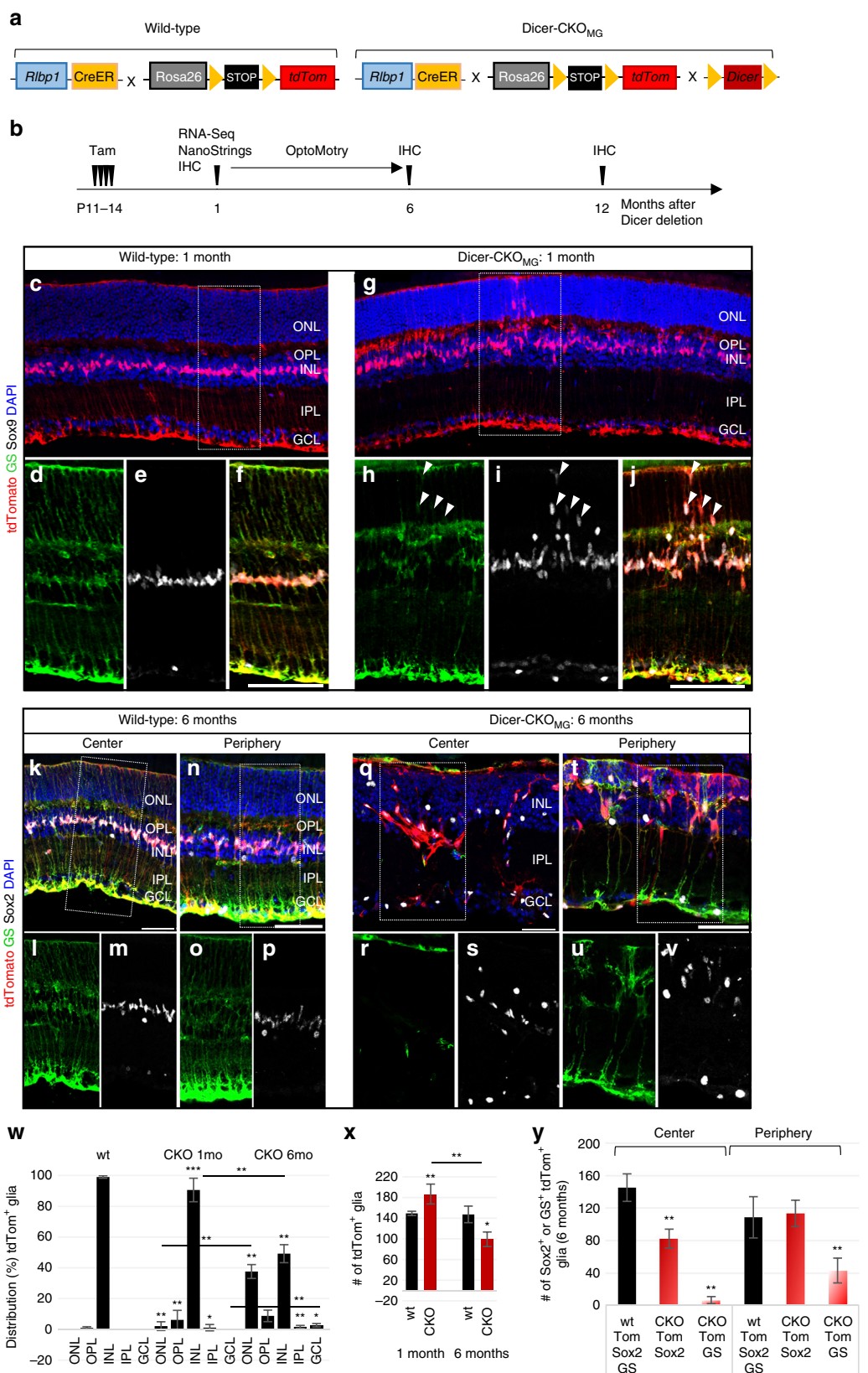

mice, but the cre-recombinase might not be as effective in the adult mouse, and/or the additional miRNAs accumulated in the adult MG[32] might decline more slowly after Dicer1 deletion.

The increase in the number of MG in retinas at 1 month after deletion might be due to an increase in proliferation of these normally quiescent cells. We failed to find evidence for ongoing proliferation in the Dicer-CKO$_{MG}$ MG after multiple EdU injections or labeling sections with markers of mitotic cells (PCNA; phospho-histone 3), and therefore conclude that the rate of MG proliferation must be very low. However, by continuous labeling of mitotic cells with EdU in retinal explant cultures from wild-type or Dicer-CKO$_{MG}$ mice (P11–P14, Supplementary Fig. 4a), we found a significant increase in the number of EdU$^+$ MG in the Dicer-CKO$_{MG}$ retinas compared to the wild-type retinas, with definite signs of cell division in the ONL (Supplementary Fig. 4b–o). Moreover, many MG in were found in the ONL, similar to what we observed in vivo (Supplementary Fig. 4p).

**Changes in gene expression after Dicer1 deletion in MG.** To determine which miRNAs might be responsible for the changes we observed in MG migration, we purified the MG by fluorescence-activated cell sorting (FACS). Retinas from *Rlbp1-creER: stop$^{f/f}$-tdTomato* were checked for successful recombination, pooled, dissociated, and FACS-purified. The fraction of the tdTomato$^+$ cells varied between 1.5 and 2.1% of all events (Supplementary Fig. 5a–d, o), in accordance with earlier reports[38]. Dicer1 heterozygous retinas showed the same pattern as wild-type retinas (Supplementary Fig. 5e–i). Consistent with the counts from retinal sections, we routinely found significantly higher fractions of tdTomato$^+$ MG in the Dicer-CKO$_{MG}$ mice compared to the wt (Supplementary Fig. 5j–o). Post-sorts of the wt or Dicer-CKO$_{MG}$ tdTomato$^+$ showed 94–96% purity of the tdTomato$^+$ cells and no positive cells in the tdTomato$^-$ fraction (Supplementary Fig. 5o).

To assess miRNAs in MG, 600 miRNAs were quantified by solution hybridization without amplification using a NanoString nCounter assay[39]. We found 23 miRNAs highly expressed in MG, including previously characterized mGliomiRs[32]. These all showed a decline in the Dicer-CKO$_{MG}$ (Fig. 4a; Supplementary Data 1), thus confirming the loss of Dicer1 function in the MG. To further validate the NanoString assay, we also used reverse transcriptase quantitative PCR (RT-qPCR) for a subset of these miRNAs to compare their levels in FACS-purified MG from the Dicer-CKO$_{MG}$ and wild-type mice. All the miRNAs we tested showed a 4–5 cycle decline in the MG from the Dicer-CKO$_{MG}$ when compared with the wild-type MG (Fig. 4b; Supplementary Data 1).

RNA-Seq was performed in order to detect changes in gene expression in FACS-purified MG between wild-type and Dicer-CKO$_{MG}$ mice, 1 month after induction (Fig. 1b). We verified the loss of exon 23 in the Dicer1-deleted MG using the RNA-Seq data. The log2 of counts per million (CPM) of the overall reads in the entire Dicer1 gene showed a reduction in the Dicer-CKO$_{MG}$

(Supplementary Fig. 5p). A Sashimi plot shows that 19 out of 32 reads skipped exon 23 in the Dicer-CKO$_{MG}$, suggesting that Dicer1 was deleted in ~60% of the cells at this time point (Supplementary Fig. 5q). That incomplete deletion could be due to MG heterogeneity[40] and might also be the reason for the observed migration of just selective groups of MG (Fig. 1g). However, even a 60% deletion is enough to lead to the severe phenotype we observed in the late phase of Dicer1 deletion.

We next analyzed the RNA-Seq data for changes in MG gene expression after Dicer1 deletion. Of all those genes expressed at a level of >5 log2 CPM, 64 genes were upregulated in the Dicer-CKO$_{MG}$ (by >2-fold, Fig. 4c; Supplementary Data 1). An analysis for enrichment in biological processes using GO Profiler showed that these 64 genes were included in developmental processes and related terms (Supplementary Fig. 6a). About 41 out of 64 genes, found in the developmental processes GO category, were analyzed using miRWalk 2.0 to look for miRNA–gene interactions in the 3′UTRs. Most of the genes had predicted and/or validated sites for five of the most highly expressed miRNAs in MG[32] (Supplementary Table 1). Despite the increase in developmental gene expression in the Dicer-CKO$_{MG}$, many glial genes (e.g., *Glul* or *Rlbp1*) were not reduced (Fig. 4c; Supplementary Table 2). This suggests that the loss of Dicer1 in MG does not lead to the loss of the glial phenotype (within 1 month), but rather the effects are more complex.

Of those messenger RNAs (mRNAs) expressed at relatively high levels, *Bcan*, showed the greatest increase (Fig. 4c, d; Supplementary Data 1). *Bcan* encodes for Brevican, a lectican family of chondroitin sulfate proteoglycans (CSPGs). Other genes that are upregulated in the Dicer-CKO$_{MG}$ include *Maff* (v-maf musculoaponeurotic fibrosarcoma oncogene family, protein F (avian)), *Atf3* (activating transcription factor 3), and *Egr3* (early growth response 3). RT-qPCR for *Bcan*, *Maff*, *Atf3*, and *Egr3* in MG from the Dicer-CKO$_{MG}$ retinas confirmed the RNA-Seq results (Fig. 4e, f, regression coefficient 0.6, Supplementary Data 1). The increase in expression of *Atf3* in the Dicer-CKO$_{MG}$ suggests that these cells were under stress, since this gene increases in expression after retinal injury[41]; however, there was no overt loss of neurons at this time in the Dicer-CKO$_{MG}$ retinas (see above), and we did not observe an increase in the reactive glial marker GFAP in the MG (Supplementary Fig. 6b–g) and therefore, it is likely, that the increase in expression of *Atf3* is due to the loss of Dicer1 and miRNAs, rather than a more general response to injury.

We focused on Brevican for further analysis, since this protein has been specifically implicated in astrocyte proliferation and migration (during development and reactive gliosis) and might therefore be involved in the MG migration that occurs after Dicer1 deletion. Interestingly, in wild-type mouse retinas, Brevican immunoreactivity was present in neurons of the GCL and the INL, but not in MG, in both, adult (Fig. 4g, h) and immature P6 retinas (Supplementary Fig. 6h–p). However, in 1 month old, as well as 1-month old Dicer-CKO$_{MG}$ mice, Brevican immunoreactivity was present in almost all MG, i.e., in

**Fig. 1** Dicer-CKO$_{MG}$ Müller glia migrate and form aggregations. **a** Schematic of the wild-type (wt): *Rlbp1CreER: stop$^{f/f}$-tdTomato* and the conditional knockout (Dicer-CKO$_{MG}$) genotype: *Dicer1$^{f/f}$: Rlbp1CreER: stop$^{f/f}$-tdTomato*. **b** Experimental design. **c–v** Immunofluorescence for tdTomato, glutamine synthetase (GS), Sox9 or Sox2, and DAPI nuclear staining of wt and Dicer-CKO$_{MG}$ retinal sections, 1 month (**c–j**) or 6 months after Dicer1 deletion (**k–v**). MG form aggregates in central retina (**q**) and display abnormal migration in the periphery (**t**). **w** Distribution (%) of tdTomato$^+$ MG in the different retinal layers in wt and Dicer-CKO$_{MG}$ retinas, wt: $n = 10$, CKO$_{1\ mo}$: $n = 8$, CKO$_{6\ mo}$: $n = 4$. **x** Numbers of tdTomato$^+$ MG per field in wt and Dicer-CKO$_{MG}$ retinas, wt$_{1\ mo}$: $n = 5$, CKO$_{1\ mo}$: $n = 9$, wt$_{6\ mo}$: $n = 4$, CKO$_{6\ mo}$: $n = 4$. **y** Numbers of Sox2$^+$ or GS$^+$ tdTomato$^+$ glia in the center and periphery of 6 months old Dicer-CKO$_{MG}$, wt$_{6\ mo}$: $n = 4$, CKO$_{6\ mo}$: $n = 4$. Statistics: mean ± S.D., Shapiro–Wilk test for normality, for **w**: Mann–Whitney U-test, for **y**: Levene's test for equality of variances and independent samples t-test, 2-tailed, as well as Bonferroni–Holm correction for all tests, significant differences: *$p < 0.05$, **$p < 0.01$, ***$p < 0.0001$. Mice of at least two different litters were analyzed. Scale bars in **f**, **j**: 100 μm, in **k**, **n**, **q**, **t**: 50 μm. GCL, ganglion cell laye; INL, inner nuclear layer; IPL, inner plexiform layer; MG, Müller glia; ONL, outer nuclear layer; OPL, outer plexiform layer; wt, wild-type

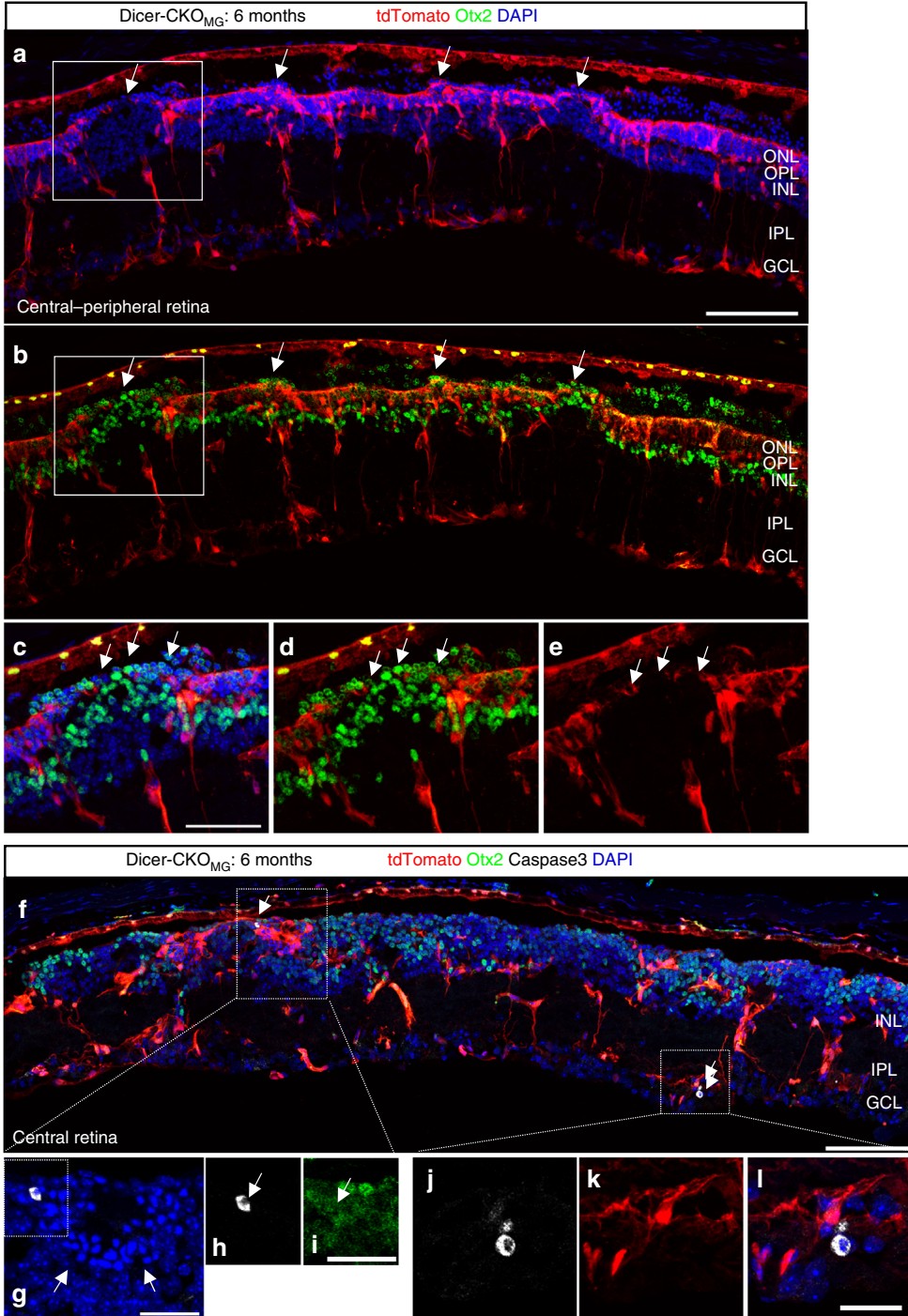

**Fig. 2** Retinal organization is disrupted 6 months after Dicer1 deletion. Immunofluorescence for tdTomato (endogenous, unlabeled), Otx2, Caspase3, and DAPI of retinal cross sections of an adult Dicer-CKO$_{MG}$ mouse, 6 months after Dicer1 deletion that received Tamoxifen at postnatal day (P) 11–14. Arrows in **a–e** show regions were the outer limiting membrane (OLM) is disrupted. Arrows in **f** show Caspase3$^+$ cells, found in the INL, in rosette like formations (**g-i**), or in the GCL (**j-k**). Mice of at least two different litters were analyzed. Scale bars in **a**, **f**: 100 µm, in **c**: 50 µm, in **g**, **i**, **l**: 25 µm. ONL, OPL, INL, IPL, GCL, as defined in Fig. 1 legend

their endfeet and in their somata (Fig. 4i–k; Supplementary Data 1). One month after Dicer1 deletion, Brevican immunor-eactivity strongly labeled MG aggregations (Fig. 4l–s).

To explore a potential role for Brevican in the abnormal migration of MG in the Dicer-CKO$_{MG}$ retina, we transduced P11 wild-type retinal MG with either Brevican-AAV (adeno-asso-ciated virus) or GFP-AAV in explant cultures (Fig. 5a). The AAV 2/1 serotype we used, almost exclusively infected MG (Fig. 5b, c). Remarkably, AAV-mediated expression of Brevican in wild-type

MG leads to an abnormal migration and aggregation of the cells similar to what we observed in the Dicer-CKO$_{MG}$ (Fig. 5d–g; Supplementary Data 1).

We also used dissociated cell cultures of adult MG to further assess the effects of Brevican expression in the Dicer-CKO$_{MG}$. MG from adult mice grow well in dissociated cell culture when plated on a "feeder layer" of immature (P12) MG (Fig. 5h). After dissociation, the primary MG cultures from Dicer-CKO$_{MG}$ mice contained more MG, compared to the wild-type cultures

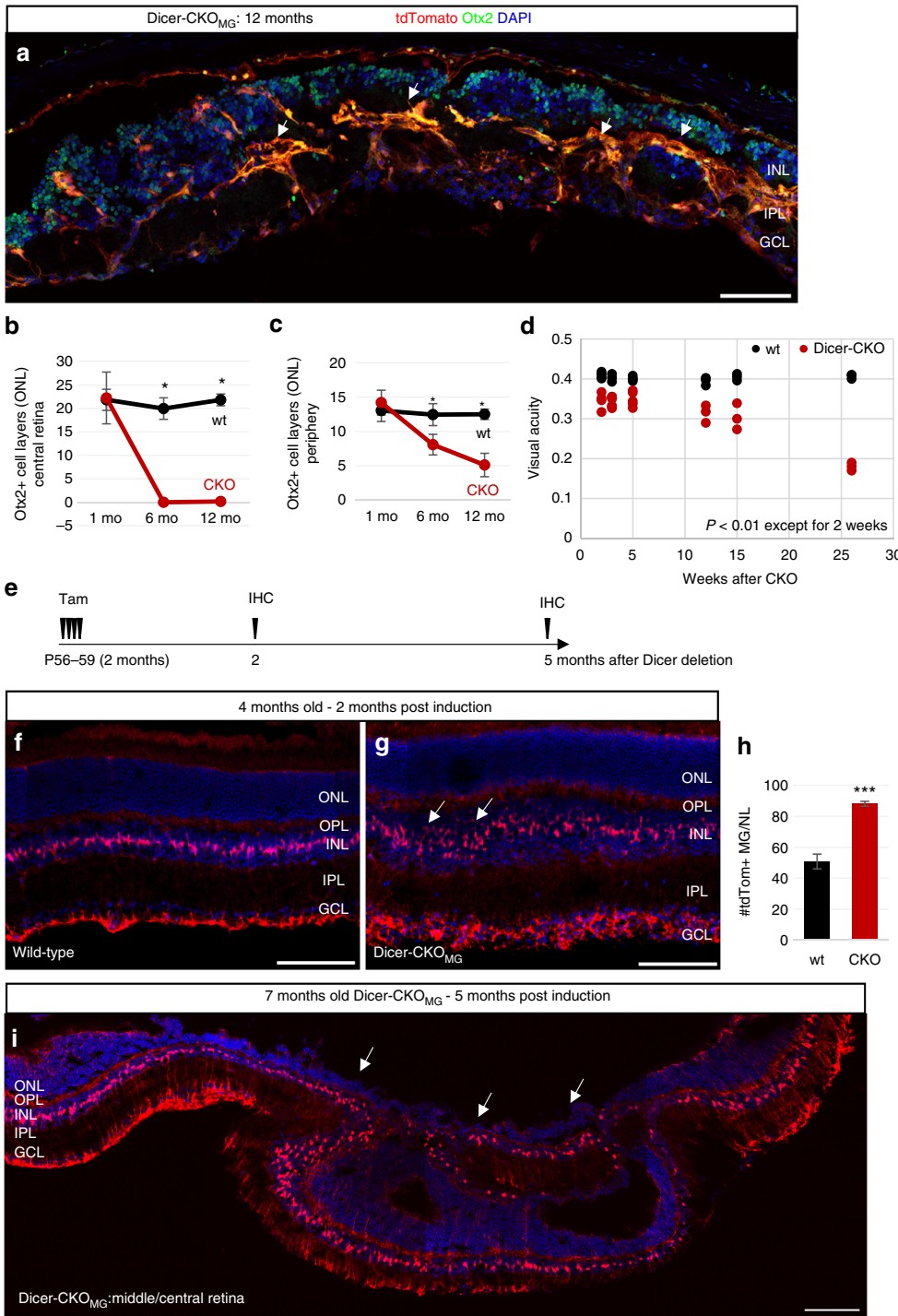

**Fig. 3** Disruption of retinal structure and function after Dicer1 deletion in MG in either juvenile or adult mice. **a** Immunofluorescence for tdTomato (endogenous, unlabeled), Otx2, and DAPI of retinal cross sections of an adult Dicer-CKO_{MG} mouse 12 months after Dicer1 deletion, which received Tamoxifen at postnatal day (P) 11–14. **b**, **c** Number of Otx2$^+$ cell layers in the ONL in the central (**b**) and peripheral retina (**c**) 1, 6, and 12 months after Dicer1 deletion in the wild-type (wt$_{1\ mo}$ $n = 3$, wt$_{6\ mo}$ $n = 4$, wt$_{12\ mo}$ $n = 3$) and Dicer-CKO_{MG} (CKO$_{1\ mo}$ $n = 4$, CKO$_{6\ mo}$ $n = 4$, CKO$_{12\ mo}$ $n = 7$). **d** OptoMotry results for wt ($n_{2\ weeks} = 4$, $n_{3\ weeks} = 3$, $n_{5\ weeks} = 4$, $n_{12\ weeks} = 3$, $n_{15\ weeks} = 3$, and $n_{26\ weeks} = 3$, one measurement per time point) and Dicer-CKO_{MG} mice ($n_{2\ weeks} = 4$, $n_{3\ weeks} = 4$, $n_{5\ weeks} = 5$, $n_{12\ weeks} = 3$, $n_{15\ weeks} = 3$, and $n_{26\ weeks}$ $n = 3$, two measurements per time point), 2–28 weeks after Dicer1 deletion ($p_{2\ weeks} = 0.000$, $p_{3\ weeks} = 0.001$, $p_{5\ weeks} = 0.002$, $p_{12\ weeks} = 0.004$, $p_{15\ weeks} = 0.007$, $p_{26\ weeks} = 0.000$). **e** Experimental design for Dicer1 deletion in the adult mouse. **f–i** Immunofluorescence for tdTomato (endogenous unlabeled) and DAPI nuclear labeling of Dicer-CKO_{MG} mice in which Dicer1 was deleted at postnatal day (P) 56–59 (2 months of age), 2 months after deletion (**f**, **g**) or 5 months after deletion (**i**). **h** Number of tdTomato$^+$ MG in the INL in wt ($n = 3$) and Dicer-CKO_{MG} mice ($n = 3$) 2 months after Dicer1 deletion. Statistics: mean ± S.D., Shapiro–Wilk test for normality, for **b**, **c**: Mann–Whitney U-test, for **d**, **h**: Levene's test for equality of variances and independent samples t-test (2-tailed), significant differences are indicated, *$p < 0.05$, **$p < 0.01$, ***$p < 0.0001$. Mice of at least two different litters were analyzed. Scale bars 100 μm. OPL, INL, IPL, GCL as defined in Fig. 1 legend

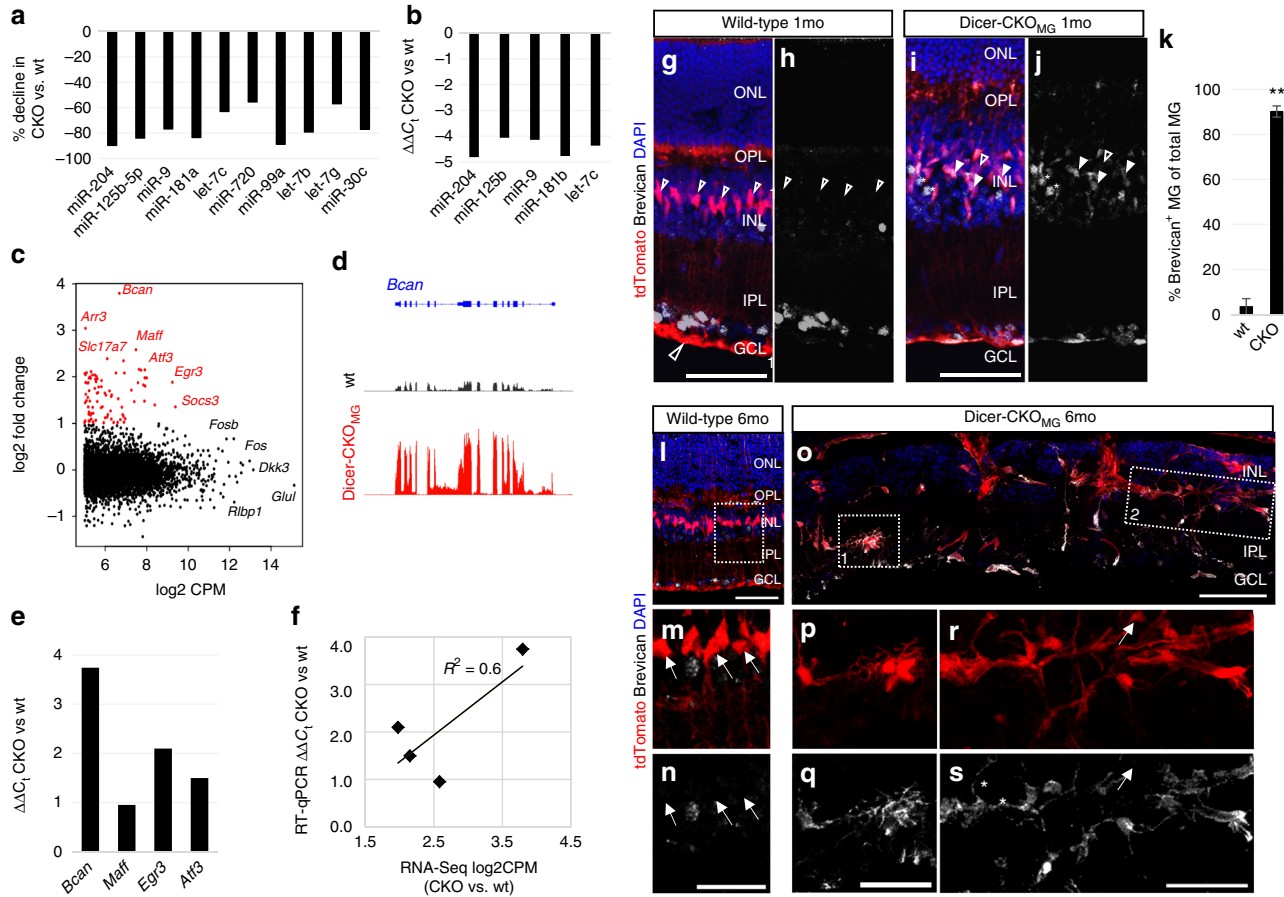

**Fig. 4** Dicer-CKO Müller glia express Brevican, which leads to disorganization. **a** Percent decline of the top 10 highly expressed MG miRNAs in the Dicer-$CKO_{MG}$, listed in order of their highest expression levels, retinas of 40 wt and 26 CKO retinas were pooled. **b** RT-qPCR comparison of the top five highly expressed miRNAs for Nanostring validation (40 wt and 26 CKO retinas). **c** Scatter plot of genes differently regulated in the MG from Dicer-$CKO_{MG}$ compared with MG from wt, shown as log2 of fold change and of counts per million (CPM). Genes showing the greatest increase are highlighted in red. **d** *Bcan* gene model (top) with the mapped reads from RNA-Seq in the wt and the Dicer-$CKO_{MG}$. **e** RT-qPCR confirmation of the top four genes upregulated in RNA-Seq in the Dicer-$CKO_{MG}$ (26 retinas, pooled) when compared to wt MG (32 retinas, pooled). **f** Comparison of $\Delta\Delta C_t$ means CKO vs. wt from RT-pPCR of four highly expressed genes in the Dicer-$CKO^{MG}$ from FAC-sorted MG (26 CKO and 32 wt retinas pooled, normalized against beta-actin) vs. log2 CPM from RNA-Seq, regression coefficient $R^2 = 0.6$. **g–s** Immunofluorescent labeling for tdTomato, Brevican, and DAPI. **g–j** 1 month after deletion: in wt mice, Brevican was expressed in neurons (asterisks) in the GCL and INL but not in MG (neither somata nor endfeet, unfilled arrows, **g, h**). Most MG in Dicer-$CKO_{MG}$ mice expressed Brevican, **i, j**, in their endfeet in the GCL and in their somata in the INL (filled arrows). Unfilled arrows point to rare MG without Brevican. **k** Percentage of MG expressing Brevican in wt ($n = 7$) and MG from Dicer-$CKO_{MG}$ mice ($n = 7$, mice of at least two different litters were analyzed). **l–s** 6 months after deletion: while the wt remains unchanged (**l–n**), tdTomato⁺Brevican⁺ MG in the Dicer-$CKO_{MG}$ form aggregates in the IPL/GCL (**o–q**) and INL/ONL (**o, r, s**). Statistics: mean ± S.D., Shapiro–Wilk test for normality and Mann–Whitney U-test, significant differences: **$p < 0.01$. Scale bars in **g, i, o**: 50 μm, in **n, q, s**: 25 μm. MG, wt, retinal layers as defined in Fig. 1 legend

(Supplementary Fig. 6q–s). After 5 days in vitro (DIV), the MG from Dicer-$CKO_{MG}$ mice started to form aggregates, while the wild-type MG remained more dispersed (Fig. 5i, j). The cells forming these aggregates had an extended morphology and these bundle-like aggregations increased over time and formed large streams of cells (Fig. 5k, l), which resembled the aggregations we found 6 months after Dicer1 deletion in vivo (Figs. 1q, 2f). To test whether the increase in Brevican expression was responsible for these aggregations, we added Brevican antibodies to the Dicer-$CKO_{MG}$ and wild-type cultures along with control IgGs. After 4 days of Brevican antibody treatment, we observed a significant reduction in the aggregation of the MG in the Dicer-$CKO_{MG}$ cultures and they now were more similar to the wild-type MG (Fig. 5m–o; Supplementary Data 1). The Dicer-$CKO_{MG}$ treated with the control IgG, however, retained the large aggregates (Supplementary Fig. 6t, u).

**miR-9 targets Brevican.** To determine whether *Bcan* is a target of the miRNAs that are highly expressed in MG, we queried the miRWalk 2.0 database (miRWalk 2.0 combines 12 target screening databases including TargetScan, miRanda, and PIC-TAR2). The analysis showed that *Bcan* is targeted by miR-9, miR-125b-5p, and let-7 (Supplementary Table 1). We compared these results of the prediction databases with CLEAR-CLIP data from P13 mouse brain, in which miRNAs, bound to their target mRNA, were directly sequenced[42]. The miRNAs targeting *Bcan* include two miRNAs that are expressed at high levels in MG: the mGliomiR miR-9[32] and miR-181a (Supplementary Fig. 6v), and these both decline substantially in the Dicer-$CKO_{MG}$ (Fig. 4a; Supplementary Data 1). These two approaches provided us with a short list of miRNAs that potentially target *Bcan*.

As noted above, the 3′UTR of *Bcan* is targeted by miR-9, miR-125b-5p, miR-181a, and let-7. To directly assess whether these

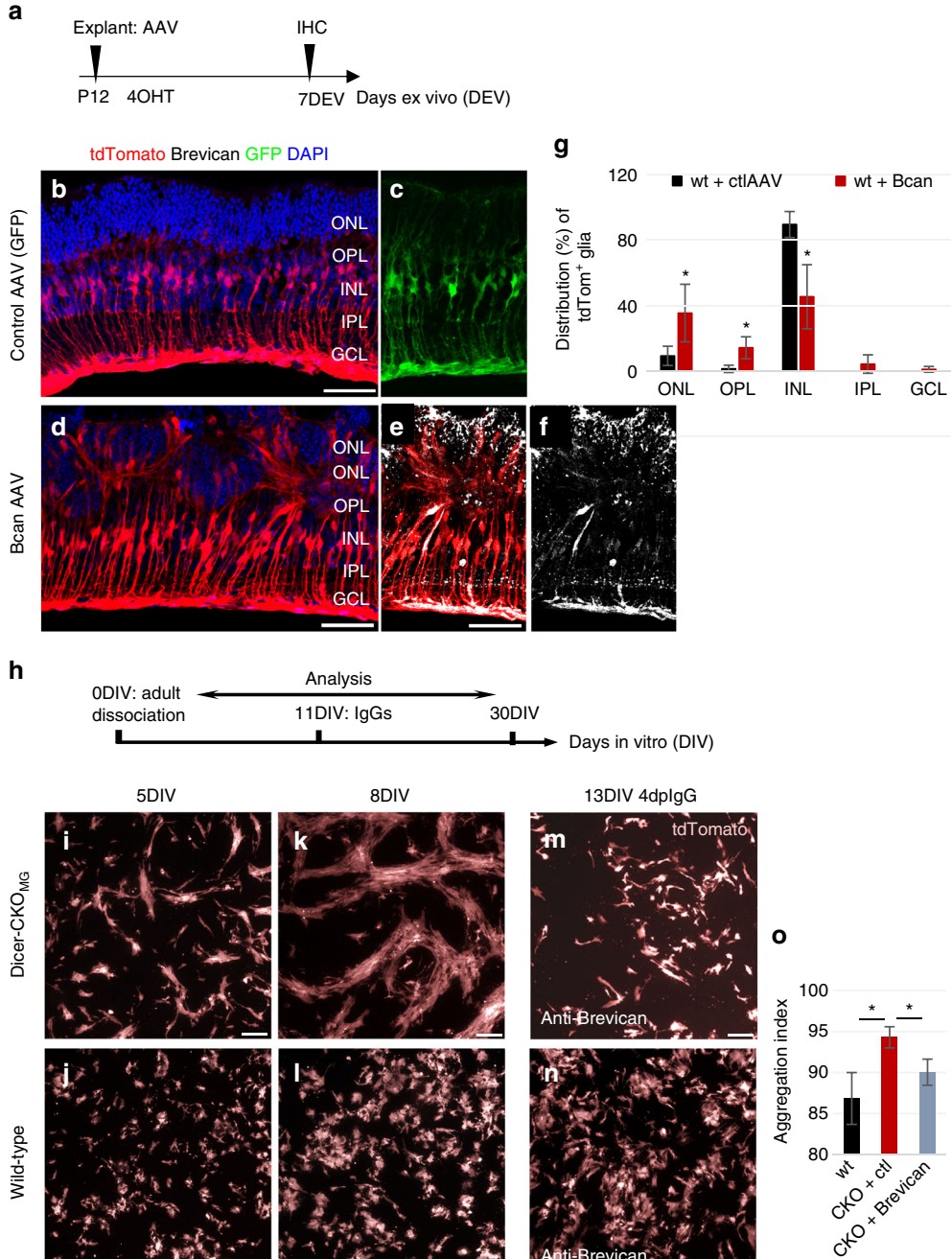

**Fig. 5** Brevican levels affect Müller glial morphology. **a** Experimental design for explant cultures. **b**–**f** P12 wild-type explants, 7 days after control-AAV (**b**, **c**) or *Bcan*-AAV transduction (**d**–**f**). Migrating, aggregation-forming MG are Brevican⁺ (**e**, **f**). **g** Distribution of MG in the retina layers after control or *Bcan*-AAV transduction, control-AAV: $n = 4$ mice, *Bcan*-AAV: $n = 4$ mice. This experiment was performed twice in the laboratory. **h** Cell culture scheme. **i**–**n** Live images of cultured tdTomato⁺ MG from adult Dicer-CKO$_{MG}$ (**i**, **k**, **m**) and wt mice (**j**, **l**, **n**). Adult MG from Dicer-CKO$_{MG}$ mice form cell aggregations over time (**i**, **k**), which are absent in the wt cultures (**j**, **l**). Brevican antibody application to Dicer-CKO$_{MG}$ cultures results in resolution of these aggregations after 4 days, but has no effect on the wt glia (**m**, **n**). **o** Quantification of the aggregation index, i.e., the percentage of areas lacking fluorescence of wt, Dicer-CKO$_{MG}$ naive, and Dicer-CKO$_{MG}$ treated with anti-Brevican, $n = 3$ mice for each condition. This experiment was performed twice in the laboratory. Statistics: mean ± S.D., Shapiro–Wilk test for normality, for **g**: Mann–Whitney U-test, for **o**: Levene's test for equality of variances and independent samples t-test (2-tailed), as well as Bonferroni–Holm correction; significant differences: *$p < 0.05$, **$p < 0.01$. Scale bars in **b**, **d**, **e**: 50 μm, in **i**–**n** 100 μm. dpIgG, days post antibody treatment; wt, ONL, OPL, INL, IPL, GCL, NFL as defined in Fig. 1 legend

miRNAs target *Bcan*, we generated a GFP sensor containing the 3′UTR of the short *Bcan* isoform (Fig. 6a) and transfected this sensor into 3T3 fibroblasts along with a mCherry transfection control. We found that miR-9 (but not miR-181a, let-7a, or miR-125b-5p) significantly reduces the GFP signal, similar to that observed from a canonical miR-9 GFP sensor (Fig. 6b–h;

Supplementary Data 1; Supplementary Fig. 7a–r). These results support a role for miR-9 in regulating *Bcan* expression in MG and suggest that the reduction in this miRNA in the Dicer-CKO$_{MG}$ underlies the phenotype we observed in vivo, ex vivo, and in vitro.

We hypothesize that miR-9 normally targets *Bcan*, in wild-type MG, and that this regulation is lost in the Dicer-CKO$_{MG}$ mice. To

test this hypothesis and possibly rescue the Dicer-CKO$_{MG}$ phenotype, we transfected adult Dicer-CKO tdTomato$^+$ MG with mimics for miR-9 or control mimics. The miRNA mimics were transfected into the MG before the onset of the cell aggregations (6–10 DIV, depending on cell growth), and the MG were allowed to survive for another week (Fig. 6i). We found that

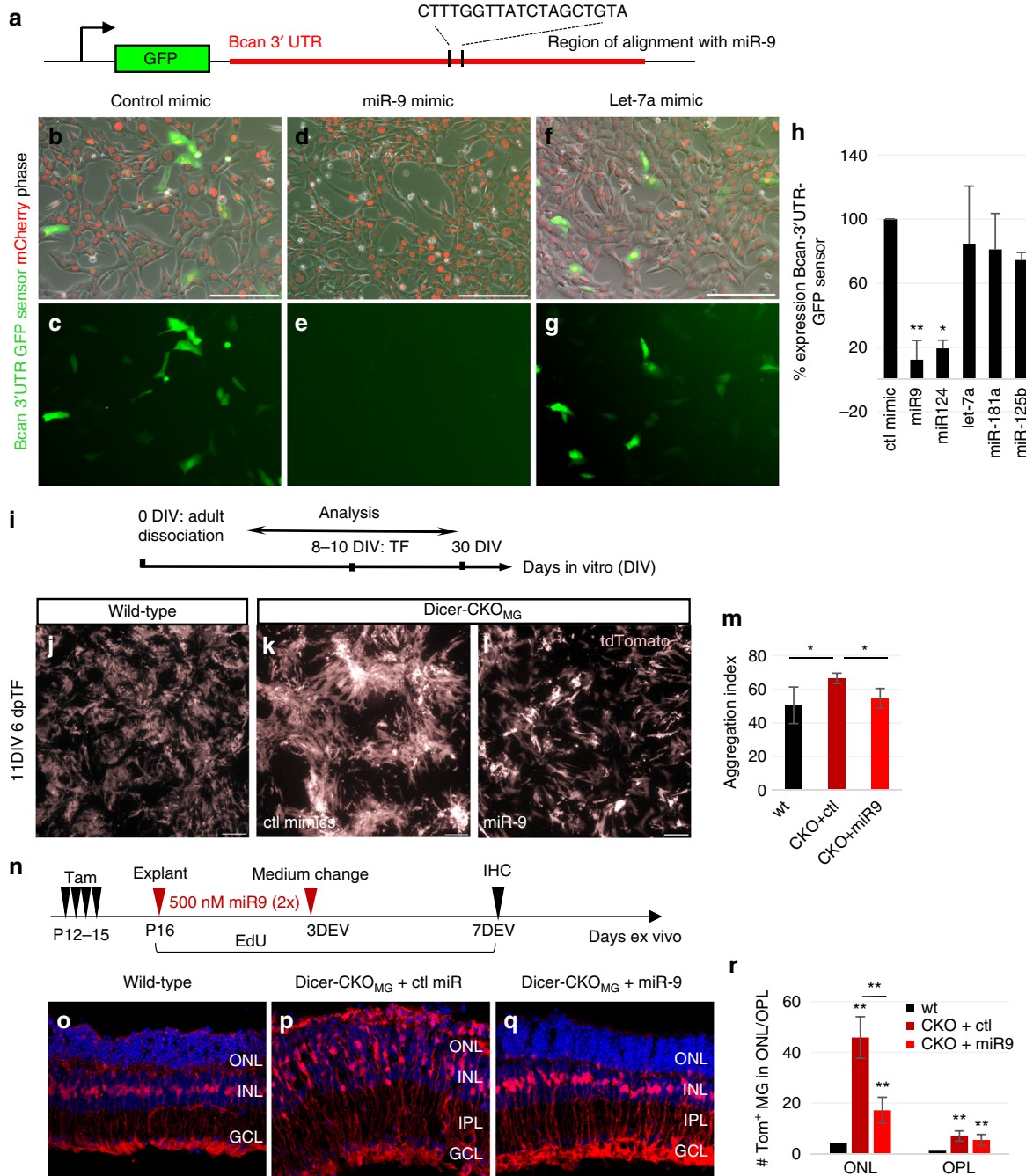

**Fig. 6** miR-9 targets Bcan and regulates Müller glial morphology. **a** Schematic of the *Bcan* 3′ UTR-GFP sensor. **b–g** Live images of 3T3 cells fibroblasts transfected with the *Bcan* 3′UTR sensor (plasmid) and mCherry mRNA transfection control with either control mimics or mimics for miR-9 or let-7a. **h** Quantification of the number of *Bcan* 3′UTR-GFP sensor$^+$ 3T3 cells of the total number of transfected cells ($n_{ctl} = 5$, $n_{miR-9} = 5$, $n_{miR-124} = 4$, $n_{let-7a} = 4$ $n_{miR-181a} = 2$, and $n_{miR-125b-5p} = 2$ individual experiments for two wells per condition per experiment). **i** Cell culture and transfection (TF) scheme. **j–l** Live images of wt MG (**j**) and Dicer-CKO$_{MG}$ cultures (**k, l**) treated with control or miR-9 mimics. **m** Quantification of the aggregation index of wt, and Dicer-CKO$_{MG}$ either transfected with control miR or miR-9, $n = 3$ mice for each condition. This experiment was performed four times in the laboratory. **n** Experimental design explant cultures. **o–q** Immunofluorescent labeling with antibodies against tdTomato and DAPI nuclear staining of P16 wt (**o**), Dicer-CKO$_{MG}$ with control mimics (**p**), and Dicer-CKO$_{MG}$ explant cultures treated with miR-9 mimics (**q**), 7 days ex vivo (DEV). **r** Numbers of tdTomato$^+$ MG per field in the ONL/OPL in wt ($n = 3$ mice) and Dicer-CKO$_{MG}$ + control miR ($n = 3$ mice) and Dicer-CKO$_{MG}$ explants treated with miR-9 mimics ($n = 3$ mice). This experiment was performed twice in the laboratory. Statistics: mean ± S.D., Shapiro–Wilk test for normality, for **h**, **r**: Mann–Whitney *U*-test, for **m**: Levene's test for equality of variances and independent samples *t*-test (2-tailed) as well as Bonferroni–Holm correction for all tests, significant differences are indicated, *$p < 0.05$, **$p < 0.01$, ***$p < 0.0001$. Scale bars 100 µm. dpTF, days post transfection; ONL, INL, GCL as defined in Fig. 1 legend

miR-9 is able to prevent the formation of the large aggregates seen in MG from the Dicer-CKO$_{MG}$ mice in vitro; miR-9-transfected Dicer-CKO$_{MG}$ MG acquired an appearance more like the wild-type MG (Fig. 6j–l) and their aggregation was reduced to that of the wild-type MG (Fig. 6m; Supplementary Data 1). This effect of miR-9 on Dicer-CKO$_{MG}$ MG was specific; we did not find the same effect for miR-181a or miR-125b-5p (Supplementary Fig. 7s–u).

We next tested whether miR-9 could prevent the glial migration that occurs in the Dicer-CKO$_{MG}$ mouse retina (Fig. 6n). We treated Dicer-CKO$_{MG}$ retinal explant cultures for 7 days with a mimic for miR-9. After miR-9 mimic treatment, most regions of the Dicer-CKO$_{MG}$ explants were less disrupted and more similar to the wt (Fig. 6o–q). Quantification of the number of MG showed that miR-9-treated Dicer-CKO$_{MG}$ explants had significantly less MG migration than the non-treated Dicer-CKO$_{MG}$ explants (Fig. 6r; Supplementary Data 1).

Previous reports have shown that in late-stage retinal degeneration (e.g., retinitis pigmentosa), the MG hypertrophy and cells migrate to abnormal positions disrupting normal retinal architecture. We examined the retinas from a small number of patients with varying levels of retinal degeneration. Consistent with earlier reports[6, 43], glial aggregations were present in the more severely degenerated retinas (Fig. 7a–c). The MG in the disorganized regions were immunoreactive for Brevican, while the MG in intact regions were not (Fig. 7d–m). Moreover, there was a significant increase of Brevican$^+$GS$^+$ co-localized pixels per area analyzed in the disturbed regions as compared to the intact regions (Fig. 7n; Supplementary Data 1); Thus establishing a correlation between an increase in this proteoglycan and the abnormal migration of MG in human retinal degeneration.

## Discussion

The importance of miRNAs for glial cell development is well-established, but less is known about miRNAs in the mature functions of these cells. We generated a conditional deletion of Dicer1 to determine the role of miRNAs in MG cells (Fig. 8). (1) We found that miRNA depletion in MG leads to disorganization of the normal retinal architecture due to MG migration and aggregation. (2) We found excessive neuronal cell death 6 months after deletion in the ONL accompanied by impairment of visual acuity. (3) RNA-Seq analysis showed, that in the Dicer-CKO$_{MG}$, the gene for the chondroitin sulfate proteoglycan Brevican (Bcan) is highly upregulated; AAV-mediated overexpression of Brevican in wild-type MG induces a phenotype quite similar to the Dicer1 deletion in MG, while blocking antibodies to Brevican prevent aggregation formation in dissociated cell cultures of MG from Dicer-CKO$_{MG}$ retinas. (4) One of the most highly expressed miRNAs in MG, miR-9, targets the 3′UTR of Brevican and treatment of Dicer-CKO$_{MG}$ MG with miR-9 mimics can rescue some aspects of the Dicer1 deletion phenotype. (5) Brevican was increased in MG in disorganized regions of retinitis pigmentosa retinas, which also display MG aggregation. Our results demonstrate the importance of miRNAs in mature MG state and function and consequently the maintenance of retinal architecture.

The loss of Dicer1 and reduction in miRNAs in MG lead to a substantial change in these cells. In a related study, Dicer1 was deleted from retinal pigmented epithelial cells (RPE) and the results of that study reported increased numbers of Caspase3$^+$ cells in the ONL, loss of pigmentation in the RPE, and increased levels of Pax6[44] 1 month after Dicer1 deletion. Since the Rlbp1CreER mice we used in this study, Cre is also active in some RPE cells[45, 46], we cannot rule out the possibility that the visual defects we observe could also be due to Dicer1 deletion in the RPE. One of the phenotypes we observed in the Dicer-CKO$_{MG}$ mice was the degeneration of photoreceptors, and this was also reported after RPE-specific Dicer1 deletion; however, other aspects of the phenotype in the Dicer-CKO$_{MG}$ mice, such as the migration of the MG and their aggregation, were not reported in the retinas of mice with an RPE Dicer1 deletion[44]. It is also unlikely that the tdTomato positive cells we find in ectopic locations in the neural retina are derived from the RPE, since all tdTomato$^+$ cells within the retina were Sox2$^+$, and this gene is not expressed in RPE cells. Moreover, we observe the same MG phenotype in vitro and ex vivo in the Dicer-CKO$_{MG}$ or Bcan-AAV treatment in the absence of RPE, further confirming the phenotype we observed is due to changes in the MG.

Deletion of Dicer1 in MG leads to an increase in their number assessed by (1) cell counts from retinal sections (in vivo), (2) FACS-purified fractions of tdTomato$^+$ cells, and (3) EdU$^+$ MG in explant cultures. Increased numbers of astrocytes have been reported previously in a GFAP-Dicer1 CKO (P7)[23]. This suggests a similar mechanism in the regulation of glial proliferation by miRNAs in the central nervous system (CNS). A recent study showed that let-7 plays a role in MG proliferation in mouse[47] and since let-7 also declines in the Dicer-CKO$_{MG}$, this could be an explanation for the increase in proliferation we observe. However, by 6 months after Dicer1 deletion in the MG, their numbers have declined to below normal, and so any increase in proliferation that occurs cannot maintain their normal numbers in these mice.

MG were displaced from their normal position in the retinas of Dicer-CKO$_{MG}$ mice. Migratory behavior somewhat like that in the Dicer-CKO$_{MG}$ was observed in some non-mammalian species after injury, when the MG re-entered the mitotic cell cycle prior to regeneration[48, 49]. This behavior was also observed in mouse retina when mutations in cell cycle inhibitor genes led to over-proliferation phenotypes[50]. In addition, long-term damage to the retina, such as that observed in slow retinal degenerative diseases, typically leads to migration of the MG from their normal positions, as the retinal structure becomes increasingly disorganized. Is the maintenance of glial position and repression of their migration dependent on miRNAs? Interestingly, an abnormal location of astrocytes has been reported in the P7 GFAP-Dicer1 CKO in astrocytes[23] and in Bergmann glia[51], consistent with our results in MG, but underlying mechanisms remained elusive.

One of the genes that increased the most in the Dicer-CKO$_{MG}$ was Brevican[52], also known as BEHAB (brain-enriched hyaluronan-binding protein)[29]. It is specifically expressed in the CNS[29, 53–55] and along with other CSPGs, such as versican or neurocan, creates perineuronal nets[56] to stabilize synapses in mature neural networks[52, 57, 58]. Interestingly, Brevican is particularly abundant in proliferating and migrating astrocytes in development, metastatic glioblastoma, and reactive gliosis[59–61]. Previous Dicer1 deletion in developing astrocytes did not report increases in Brevican expression. One possible reason for this difference in Brevican expression between these two types of glia could be miR-9. Even though miR-9 has been reported to be expressed by other glial types in the nervous system, including young astrocytes[23], the levels in the MG are considerably higher than those in mature astrocytes. Thus, it is possible that Brevican and miR-9 are involved in the different responses to injury observed between MG and astrocytes.

Since not much is known about Brevican, it is possible that Brevican not only causes MG aggregation but also a loss in cell polarity of the MG. One of the first features of the Dicer1 deletion in the MG is a disruption in the OLM. The loss in junctional complexes at the apical side of the cells may lead to an epithelial–mesenchymal transition and cause the cells to migrate through the retina. Another interesting feature of the loss of Dicer1 in the MG is the increase in retinal thickness.

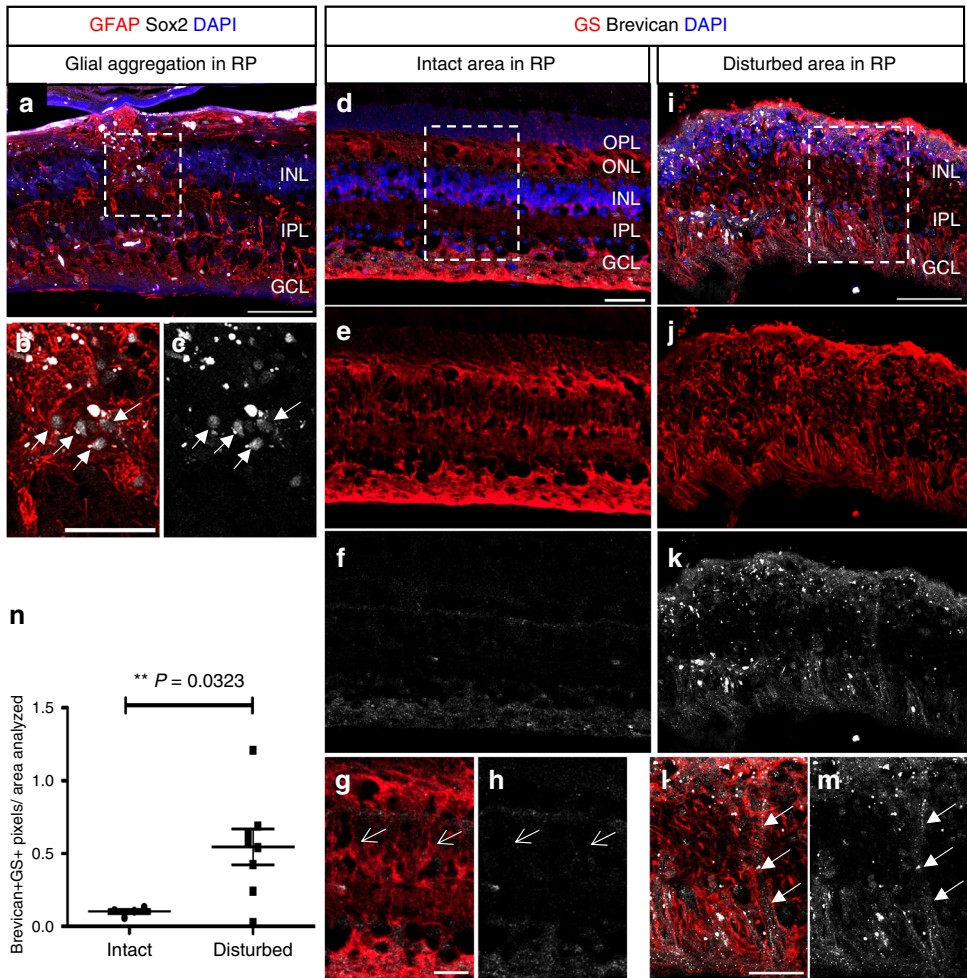

**Fig. 7** Brevican in MG of disorganized human retinitis pigmentosa retinas. **a–m** Immunofluorescence for Tomato, GFAP, and Sox2 (**a–c**) or glutamine synthetase (GS) and Brevican (**d–m**), as well as DAPI nuclear staining of retinal cross sections of retinitis pigmentosa patients. Glial aggregations (white arrows in **b**, **c**) found in the INL resemble to those found in the 6-month-old Dicer-CKO$_{MG}$ retinas. MG in intact regions (arrows in **g**, **h**) were Brevican$^-$ while glial aggregations found in disturbed regions (arrows in **l**, **m**) were Brevican$^+$. **n** Brevican/GS immunoreactivity assessed in intact ($n = 4$) and disturbed ($n = 8$) retinitis pigmentosa samples showed a significant increase in Brevican protein level in the disturbed samples. Statistics: mean $\pm$ S.E.M., independent samples $t$-test and Levene's test for equality of variances, 2-tailed, $p = 0.0323$. Scale bars in **a**, **i**: 100 μm, in **b**, **d**: 50 μm in **g**, **l**: 25 μm. ONL, OPL, INL, IPL, GCL, NFL as defined in Fig. 1 legend, RP retinitis pigmentosa

Interestingly, a similar stretching phenomenon has been reported when MG are ablated and the authors propose that MG act like springs to hold the tissue together and ensure the tensile strength of the retina[4]. Perhaps the Dicer1-depleted MG lose their tensile strength along with their polarity. Moreover, after long survival periods post-Dicer1 deletion, there was significant thinning of the ONL and severe retinal disorganization, reminiscent of previous studies using MG ablation[3, 62]. The phenotype we observed after Dicer1 deletion thus raises a number of interesting questions of the role of miRNAs in MG cell biology in conjunction with progressive retinal degenerative disease including neuronal and vascular pathologies that will require further investigation.

Taken together, our data show that Dicer1 and miRNAs are important to maintain MG in their normal position in the retina, and to support normal retinal architecture and function. Loss of miRNAs in the MG after Dicer1 deletion led to a small increase in mRNA levels of many genes, while a few genes showed large increases in their expression levels. The increase in the level of *Bcan*/Brevican expression in the Dicer-CKO$_{MG}$ led to changes in their aggregation behavior in vivo, ex vivo, and in vitro and potentially an increased migratory behavior in vivo. Thus, normal levels of miRNAs, in particular of miR-9, are clearly important for

the maintenance of the stable MG state, and raise the possibility that changes in miRNAs might be related to the remodeling of retinal structure, and that occurs in long-term retinal degenerative diseases. Therefore applications of miRNAs could help to overcome the limitations of current attempts to restore retinal function by transplantation, gene therapy, or prosthetic devices to treat long-term retinal degenerative diseases.

## Methods

**Animals**. All mice were housed at the University of Washington, Department of Comparative Medicine and procedures were approved by the University of Washington Institutional Animal Care and Use Committee (UW-IACUC). *Rlbp1CreERT2* mice were crossed to *R26-stop-flox-CAG-tdTomato* mice (Jackson Labs, also known as Ai14; 129SvJ background) and will be henceforth referred to as *Rlbp1CreER: stop^{f/f}-tdTomato* or wt. For the Dicer1 conditional knockout, *Rlbp1CreER: stop^{f/f}-tdTomato* mice were crossed to *Dicer^{f/f}* mice (Jackson Labs; 129SvJ background), *Dicer^{f/f}: Rlbp1CreER: stop^{f/f}-tdTomato* are referred to as the conditional knockout (Dicer-CKO$_{MG}$, CKO). Littermates that were heterozygous for the floxed Dicer1 allele were also used for tissue analysis, and showed no difference with the wild-type phenotype. For the progenitor reporter mouse, *stop^{f/f}-tdTomato* mice were crossed to *Sox2CreER* mice (Jackson Labs). For lineage, trace the *Nrl-GFP: Rlbp1CreER: stop^{f/f}-tdTomato* mouse was used as reported before[32]. Tamoxifen (Sigma, St. Louis, MO) was administered intraperitoneally at 75 mg kg$^{-1}$ in corn oil at P11–14 or P56–59 to initiate the recombination of the floxed alleles. The thymidine analog 5-ethynyl-2′deoxyuridine (EdU, Invitrogen, 1 μM)

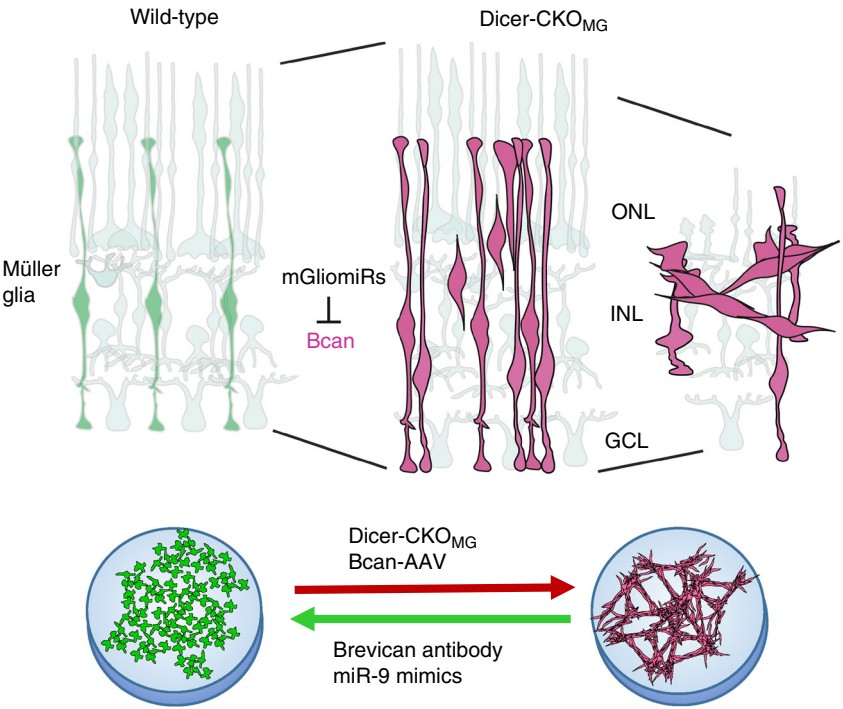

**Fig. 8** Suggested mechanism for microRNA regulation of Müller glia state. miRNA depletion in retinas and cultures of MG in the Dicer-CKO<sub>MG</sub> mice leads to disorganization of the normal retinal architecture due to MG migration and aggregation, in vivo and in vitro. The gene upregulated the most in the Dicer-CKO<sub>MG</sub> and responsible for MG migration and aggregation is *Bcan*, which encodes for the chondroitin sulfate proteoglycan Brevican. AAV-mediated overexpression of Brevican in wild-type MG also leads to MG migration and aggregation while blocking antibodies to Brevican prevent MG aggregation in dissociated MG cultures from Dicer-CKO<sub>MG</sub> retinas. miR-9, one of the most highly expressed miRNAs in MG (mGliomiRs), targets the 3′UTR of Brevican and miR-9 mimics treatment of Dicer-CKO<sub>MG</sub> MG can prevent MG aggregation and restore normal MG shape in vitro

was injected daily up to 3 weeks to detect MG proliferation. For all individual experiments, a sample size of at least three mice per condition were chosen to ensure adequate power to detect a pre-specified effect size. No animals were excluded. Both, males and females were used. All mice were randomly assigned to treatment groups/ experiments. Age of Tamoxifen administration and subsequent analysis as detailed in the text.

**Fluorescence-activated cell sorting**. Prior to FACS, all retinas were checked for successful recombination under the fluorescence microscope. For each sort, 6–10 retinas were pooled and dissociated in DNase/Papain (75 µl/750 µl respectively, Worthington) for 20 min at 37 °C on the shaker, triturated, mixed with Ovomucoid (750 µl), centrifuged for 10 min at 300×g and resuspended in 800 µl Dnase/Ovomucoid/Neurobasal solution (1:1:10 respectively, Gibco) per retina. Before sorting, cells were filtered through a 35 µm filter and sorted using an 85 micron nozzle and collected into two separate chilled tubes, one for the tdTomato⁺ MG and one for the tdTomato⁻ neurons. Debris was excluded from the sort and only all events in gate P1 were sorted (Supplementary Fig. 5a, e, j). Cells with the brightest fluorescence were found in gate P3 ("positives", MG fraction), cells with no fluorescence in gate P2 ("negatives", neuronal fraction, Supplementary Fig. 5b, f, k), everything in between was excluded. Samples were collected in FBS-coated tubes containing neurobasal medium. After collection, the tdTomato⁺ MG fraction (P3) and the tdTomato⁻ fraction (P2) were post-sorted to validate purity. Cell sorts were performed using BD Aria III cell sorter (BD Bioscience). After collection, the tdTomato⁺ MG fraction and the tdTomato⁻ fraction were post-sorted to validate purity. In addition, one drop of each condition was plated on a coverslip and evaluated for purity. All other cells were spun for 10 min at 300×g at 4 °C, the pellet was homogenized in Qiazol and stored at −80 °C (Qiagen).

**Plasmids and viral production**. *Bcan*-AAV construct was obtained from Dr M.C. van den Oever, University of Amsterdam[63], the GFP-AAV control construct serotype 2/1 was purchased from Addgene and packaged into adenoviral vectors using the helper plasmid pDP1rs (1 mg ml⁻¹, Plasmid Factory; Bielefeld, Germany). The AAV and helper plasmids were transfected into HEK cells with CaCl₂ and the media harvested after 3 days. Virus was concentrated by ultracentrifugation for 90 min at 25,000×g at 4 °C. The concentrated virus was resuspended in media and either used directly for retinal infection or frozen in aliquots for later use. The 3′ UTR *Bcan* short version sensor was synthesized by Genecopoeia (Rockville, MD, USA), and was used for DNA transfection as described below.

**Müller glia co-culture**. In order to culture adult MG, we established a co-culture method using MG from postnatal day (P) 11/12 mice. For the "feeder layer", MG were dissociated (see above) and grown in neurobasal medium with N2 supplement, tetracycline-free 10% fetal bovine serum (FBS, Clontech), and epidermal growth factor (EGF, R&D Systems, 100 ng ml⁻¹). After 6–7 DIV, cells were passaged and 1 day later used as a feeder layer for adult FAC-sorted glia or adult glia. MG co-cultures were either maintained in 6-well plates or on Poly-D-lysine (PDL) and Matrigel (Corning) covered coverslips in 24-well plates. EdU (10 µg ml⁻¹, Invitrogen) was added to track cell proliferation. Cultures which did not show appropriate cell growth were excluded from further procedures.

**Antibody treatment and transfection**. *Antibody treatment*: After 8–12 days in vitro, we treated cultures of MG from *Rlbp1CreER: stop^{f/f}-tdTomato* or *Dicer^{f/f}: Rlbp1CreER: stop^{f/f}-tdTomato* mice with either rabbit anti-Brevican antibodies (40 µg ml⁻¹, Supplementary Table 3) or control rabbit anti IgGs (40 µg ml⁻¹) in normal medium for 4 consecutive days.

*MG transfection*: We used miRNA mimics (Supplementary Table 4, Thermo Scientific; 500 nM each) for RNA transfection. MG cultures from Dicer-CKO<sub>MG</sub> and wild-type mice were transfected using Lipofectamine 3000 in Optimem medium, in accordance with manufacturer's instructions. About 3 h after transfection, the media was changed to normal medium (10% FBS with EGF and EdU).

**3T3 fibroblast cultures and transfection**. 3T3 fibroblast were purchased from ATCC and grown in DMEM medium with tetracycline-free 10% FBS (Clontech). For transfection we used the *Bcan* 3′UTR plasmid or miR-9 GFP sensors (1 mg ml⁻¹) along with miRNA mimics (Thermo Scientific; 500 nM each, Supplementary Table 4) and in combination with mCherry RNA. To generate mRNA for transfection, we used the pSLU plasmid[64] and the Ambion mMessage mMachine (AM1345; Ambion) according to manufacturer's instructions.

**Explant culture**. After killing, the retinas from P15-P22 old mice were isolated in cold HBSS, mounted on a 0.4 µm pore tissue culture filter insert and cultured, with one retina/well of a 6-well plate for 1 week in DMEM medium with N2 (Insulin, Transferrin, Progesterone, Putrescine, Selenite), 1 % dialyzed FBS, 4-hydroxytamoxifen (4-OHT; 1:6000, Sigma), and EdU (10 µg ml⁻¹, Invitrogen) at the gas–liquid interface[65]. miRNA mimics (500 nM) were added directly to the

medium. AAV viral particles (AAV-GFP serotype, 2:1; or AAV-*Bcan*) were applied on top of the explant. Explants which showed substantial cell death due to the culture procedures were excluded from evaluation.

**Immunofluorescent labeling**. Eyes or explants from mice were fixed in 4% paraformaldehyde (PFA) for 30–60 min, treated with 30% sucrose in PBS overnight, embedded in O.C.T. embedding medium and cross sectioned in 12 μm thick sections. MG co-cultures were fixed with 2% PFA. Human retinal sections of retinitis pigmentosa patients were obtained without identifiers from Dr. Ann Milam (University of Washington)[66]. For immunofluorescent staining, cells were incubated in blocking solution (5% milk block: 2.5 g nonfat milk powder in 50 ml PBS; with 0.5% Triton-X100) for 1 h at RT. Primary antibodies (Supplementary Table 3) were incubated in 5% milk block overnight, secondary antibodies (Invitrogen/ Molecular Probes and Jackson ImmunoResearch, 1:500–1000) for 1 h at RT and counterstained with 4′,6-diamidino-2-phenylindole (DAPI, Sigma, 1:1000). EdU labeling was carried out using Click-iT EdU Kit (Invitrogen).

**NanoString and reverse transcriptase quantitative PCR**. The sorts of 26–40 retinas were pooled for the RNA purification. RNA was extracted and purified with a miRNeasy Micro Kit in accordance with manufacturer's instructions (Qiagen). RT-qPCR was performed using SsoFast EvaGreen Supermix (Bio-Rad) on a Bio-Rad Thermocycler. For mRNA analysis, complimentary DNA (cDNA) was synthetized using the iScript cDNA Synthesis Kit (Bio-Rad). Primers are shown in Supplementary Table 5. For miRNA RT-qPCR, cDNA stem-loop RT primers were used to produce cDNA for specific miRNAs[67, 68]. Primers are shown in Supplementary Table 6. RT-qPCR was performed using SsoFast EvaGreen Supermix (Bio-Rad) on a Bio-Rad Thermocycler. Reactions were run in triplicates or duplicates for RT or –RT samples. Values were normalized to 5sRNA and/or β-actin. Delta delta $C_t$ between Dicer-CKO$_{MG}$ and wt were calculated and were expressed together with the standard deviation (S.D.). NanoString nCounter was used for miRNA expression analysis. About 200 ng total RNA per sample (33 ng μl$^{-1}$) was submitted for NanoString analysis, performed at Fred Hutchinson Cancer Center in Seattle, WA, USA. NanoString data was analyzed using nSolver 2.6 software. The data represent counts of molecules normalized against four housekeeping genes (β-actin, *GAPDH*, *Rpl19*, and *B2m*), eight negative controls, and six positive controls that were run with the samples.

**RNA sequencing**. For RNA-Seq 500 ng per sample (50 ng μl$^{-1}$) was sequenced on an Illumina HiSeq and reads that passed Illumina's base call quality filter were mapped to mm10 using TopHat v2.0.12. To generate counts for each gene using htSeq-count v0.6.1p1, in "intersection-strict" overlap mode, genes with zero counts across all samples were removed, and data normalized using edgeR v3.12.0. Further analysis was done using Bioconductor and R (version 3.2.3).

**Analysis of miRNA targets and alignments**. miRBase (www.mirbase.org) was used to get the miRNA sequences, Serial Cloner 2.6 was used to find alignments. For target gene prediction miRNAs, we used the databases from miRWalk 2.0, a comprehensive atlas of predicted and validated miRNA–target interactions that combines the leading prediction databases (http://zmf.umm.uni-heidelberg.de/ apps/zmf/mirwalk2/custom.html). We further used the University of California Santa Cruz (UCSC) genome bioinformatics browser (http://genome.ucsc.edu) to analyze miRNA binding sites using CLEAR-Clip data provided by Dr M. Moore (Rockefeller University)[42].

**Microscopy and statistical analysis**. Live imaging was performed using Zeiss Observer D1 with Axio-Cam. Fixed cells were analyzed by Olympus FV1000 or Zeiss LSM 880 confocal microscope. For cell cultures, six random fields per coverslip, at ×200 magnification were counted for every condition. For retinal cross sections, 2 areas per section with 625 × 625 μm dimension at ×200 magnification, four optical sections of 2 μm thickness, for five sections per mouse, of at least eight Dicer-CKO$_{MG}$ mice, six wild-type mice, and six Dicer1 heterozygous (hets) were counted. The exact *n* is given in the particular figure legend. For the INL measurements, areas with 200 × 200 μm dimension at ×600 magnification, four optical sections of 2 μm thickness were analyzed. For cell cultures, live images were taken at ×100 magnification and at least six images of at least three individual Dicer-CKO$_{MG}$ or wild-type mice were counted or analyzed by ImageJ. Values are expressed as mean ± S.D. Statistical analyses were performed by Shapiro–Wilk test for normality. In case of no proof of normal distribution, the Mann–Whitney *U*-test was performed, in case of normal distribution, Levene's test for equality of variances and independent samples *t*-test, 2-tailed were performed using IBM SPSS Statistics 19 program. The Holm–Bonferroni method was used to correct for multiple comparisons. Cell counts were performed on samples containing the mouse number but no phenotype identifier.

**Visual acuity testing**. Visual acuity was tested by the Optomotry system[37]. Briefly, freely moving mice were exposed to moving gratings of various spatial frequencies (cycle per degree) at full contrast using the simple staircase method. Parameters: contrast 100%, drift speed 012.0 d s$^{-1}$, temporary frequency 03.8 Hz, screen width

40.6 cm, screen distance 23.5 cm, max. frequency 0.75 cycle per degree, angle 81.6°, radius 46.9 cm. Mice were tested 2, 3, 5, 12, and 15 weeks after Dicer1 deletion and corresponding controls, $n \geq 3$, with two measurements per time point for the Dicer-CKO$_{MG}$ mice, one measurement per time point for wt mice. For wt measurements, a different group of mice were analyzed at the particular time point to confirm baseline of 0.4 cycles per degrees and to increase the number of biological replicates (visual acuity does not change starting at P25)[37]. For the Dicer-CKO$_{MG}$ mice, one group of 5 mice were tested 2, 3, and 5 weeks after Dicer1 deletion, and another group of 3 mice were tested 12, 5, and 26 weeks of Dicer1 deletion. Visual acuity testing was approved by IACUC, protocol number 2448-08.

**Analysis of MG aggregation**. All tissues that were directly compared were stained in a single batch and live imaging or confocal settings were identical for each captured image. To quantify the degree of aggregation of the MG in dissociated cultures, we quantified the inverse of the area of the plate covered by the cells using ImageJ (NIH, Bethesda). Images were converted to 8-bit, contrast enhanced by CLAHE, and converted to a binary image by applying an identical threshold to all images. Aggregates in the binary images were then quantified using the "Analyze Particles" tool. To assess the fluorescent intensity of Brevican immunoreactivity in the explants, the mean gray value within tissue sections of the same field used to analyze the rescue of MG migration were analyzed using ImageJ (NIH, Bethesda). To assess co-localization of the Brevican signal within the MG, images were analyzed in ImageJ (NIH, Bethesda). Single plain images of retinal sections collected from individuals with retinal disease were used to determine if Brevican was expressed in areas of retinal disruption compared to relatively intact regions of the same retinas. Sections were stained with Brevican and GS. A region of interest was drawn around the retina using the "Polygon selections" tool and the area of this analyzed region was recorded. Images were then converted to 8-bit and the gray value of each pixel in the GS and Brevican channel was compared on a scatter plot to assess co-localized pixels. Any pixel having gray values >25 in both channels were considered co-localized and counted. Graphs were generated showing the number of co-localized pixels normalized to the area analyzed. All analysis was performed by a lab member blinded to the conditions.

**Data availability**. Nanostrings and mRNA-Seq data have been deposited in GEO (PRJNA400179 GSE103098) and the SRA (NCBI, SRP115835) respectively.

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

## Acknowledgements

We thank the Peter Rabinovitch lab for assistance with the FAC-sorter. The Fred Hutchinson Cancer Research Center for the RNA service, Dr M.C. van den Oever, University of Amsterdam, for the *Bcan*-AAV construct, Ellen Bercaw, Olivia Bates, and Jessica Tondo for technical assistance, Dr Olivia Bermingham-McDonogh for the Brevican antibody suggestion and manuscript edits, and the Reh lab for helpful

manuscript comments. Grants/financial support provided by the National Eye Institute grant NEI R01EY021482 to T.A.R. and scholarship Wo 2010/1-1 for S.G.W. from Deutsche Forschungsgemeinschaft (DFG). Grant #TA-RM-0614-0650-UWA from the Foundation Fighting Blindness. The Vision Core Grant P30EY01730 for use of the imaging facilities.

## Author contributions

S.G.W. and T.A.R. designed and conceived the study, and S.G.W. performed the experiments and data collection. N.L.J. developed the macros for fluorescence intensity and aggregation measurements, and analyzed the images. E.M.L. provided the *Rlbp1CreER* mouse. S.G.W. and T.A.R. analyzed the data and wrote the manuscript.

## Additional information

**Competing interests:** The authors declare no competing financial interests.

