## [Peer Review File · Nature Communications]

Reviewers' Comments:

Reviewer #1:

In this paper, the authors use a conditional knockout of Dicer in MG and find that loss of Dicer leads to abnormal migration and ultimately aggregation of MG. They identify the most down regulated miRNAs after Dicer deletion and then find that loss of miR-9 leads to up regulation of Brevican. The experiments in general are solid, the drawback is that the paper is largely descriptive, observational with little mechanistic insight into what role Brevican might play.

1. The description of the mice is not cited or written correctly. This was a problem in a previous paper yet the authors make the same mistakes in this paper.
2. Why do only some MG respond with significant abnormal migration?
3. What happens if Dicer is deleted in adult MG?
4. How extensive is cell death upon loss of Dicer?

Reviewer #2:

This paper presents some very interesting data on the role of miRNA in maintaining retinal Muller glia (MG) homeostasis and retinal architecture. The authors make use of an inducible Cre reporter line to examine the consequences of deletion of Dicer1 on retinal histology, gene expression (NanoSting nCounter assay for miRNA and RNA-Seq), and function, using a variety of in vivo and in vitro approaches. They provide data suggesting that much of the retinal abnormality that they observe is due to overexpression of the proteoglycan Brevican, and that Brevican overexpression is due to decreased expression of miR-9.

Overall the manuscript is well written and provides significant new and interesting data that should be of interest to the Nature Communications readership, of interest not only to the retinal community but also to the more general glia/neuroscience communities. Despite the strengths of the manuscripts, below are some suggestions to clarify, add depth, and expand the presented work. Although clearly all the suggestions need not be taken, if the authors could address the ones that they feel most relevant to their work this could significantly increase the significance and interest of the work to the NC readership.

1. The work by Vázquez-Chona et al, which presumably describes the same promoter used by the authors, indicates that the Rlbp1 promoter is not totally MG-specific. This is probably not an issue for interpretation of the presented data, but should be acknowledged since it could theoretically contribute to the observed phenotype.
2. The authors state that in the induced Dicer-CKOMG retinas mice there is not only abnormal MG migration but also that the retinas show a 45% increase in INL thickness. However, the increase in tdTomato+ MG cells is only 23%, and MG make up only a fraction of the INL. This seems like an important finding and it would be helpful if the authors could describe in more detail the cellular basis of the INL thickening and at least discuss possible

mechanisms.

3. More exploration of the mechanisms of the expanded MG population. The authors mention increased proliferation as a likely possibility, but limited EdU experiments did not confirm this. More extensive exploration would be helpful. Since the phenotype is evident in explants, live imaging with time-lapse analysis could be very useful. Are results that they have observed different between explants and in vivo, as perhaps suggested by the text?

4. It is interesting that abnormalities in visual function, as measured by OptoMotry®, were evident as early as 2 weeks following miRNA depletion. Yet most of the molecular studies were performed at one month post Cre induction. Although of course all the molecular studies do not need to be repeated, since early time points could point more towards causative changes, as opposed to secondary changes, it would be helpful for the authors to test some of the miRNA and transcriptomic changes noted by their genome-wide assays at earlier time points. Looking at at least some additional timepoints would be an important addition to the paper and should be done.

5. What was the actual level of Dicer1 expression in the purified induced MG cells? This seems important to confirm the KD, and this data should already be available from the RNA-Seq data, and would be good to also test by QPCR (sorry if this data is already presented and I missed it?)>

6. Are all the primary datasets available in a publicly accessible database. This seems important, and presumably is a requirement of NC?

7. It seems that some of the RHA-Seq data findings were confirmed by QPCR. Would be helpful to give more details. Were the samples tested by QPCR the same as the ones used for RNA-Seq, or were they independent samples – the latter would of course be better. How many replicates were tested? A scatter plot to compare fidelity between RNA-Seq and all QPCR data as a supplementary figure would be useful.

8. The data on glia aggregates and Brevican expression in retinitis pigmentosa retinas is potentially very interesting and important. But no details are provided – how many individual patient retinas were tested, what were the patient phenotypes, genotypes, how does the staining compare to control age-matched retinas. Although this work potentially adds significantly to the manuscript, if it can not be presented in more detail than it might be better not to present it at all.

9. In the Discussion the authors state, “We found that miRNA depletion in MG leads to disorganization of the normal retinal architecture and ultimate impairment of visual acuity due to MG migration and aggregation.” Although the authors have clearly demonstrated retinal disorganization and MG migration and aggregation, I am not sure that they have experimentally shown that the disorganization and decreased visual acuity is mechanistically DUE TO the MG migration and aggregation.

In summary, a potentially interesting and important manuscript, but it would benefit from

more detailed and in depth analysis.

Reviewer #3:

In their manuscript titled "Muller glial microRNA are required for the maintenance of glial homeostasis and retinal architecture" the authors conditionally inactivate Dicer1 in mature muller glia. They observed proliferation, migration of the cell nucleus to the apical region of the retina and changes in miRNA and gene expression. Importantly, they did not observe upregulation of GFAP suggesting that at least some features of reactive gliosis were not present in these cells. The authors then focused on Brevican which was one of the most upregulated genes in the cells. They found that AAV mediated delivery of Brevican led to a similar phenotype as Dicer deletion in Muller glia. Next ,they performed in vitro studies on primary MG cultures with dicer knockout cells and Brevican neutralizing antibody. They showed aggregation in the Dicer deficient MG in this system and Brevican antibody blocked this effect. Next, the authors analyzed the miRNA targets with an emphasis on Bcan. They found that miR-9 was important for Bcan expression and was able to rescue some aspects of the Dicer deletion phenotype. Overall, the authors show solid data linking miR9 to Bcan and the muller glia phenotype. The studies are well executed and the appropriate experimental controls are performed. The only limitation of the study is the relevance to disease or physiology. The reduction in vision is important but this is not surprising given the retinal disruption. A more direct connection to human retinal disease would strengthen the manuscript.

Reviewer #4:

Unclear to me why the authors elect to delete Dicer with Tam injections at P11-P14. If the argument is that miRNAs are required for maintenance of mature Muller glia structure and function, I would like to see the effects of conditional deletion once animals reach adulthood. This also applies to the retinal explant culture experiments where the authors used P11 animals.

Evaluation of overall retinal architecture by histology should be performed. Previous descriptive studies following Muller glia ablation have described rosette formation, which the aggregates by IHC seem to suggest.

The link between Muller glia and retinitis pigmentosa is glossed over and needs clarification. Are glia in control age-matched patient samples also immunoreactive for brevican?

Furthermore, the claim in the Abstract that "MG from RP patients display an increase in Brevican immunoreactivity at sites of MG aggregation, linking the retinal remodeling that occurs in chronic disease with miRNAs" is overstated. Figure 5 only shows that glia in RP retinas express Brevican, and does not demonstrate that brevican immunoreactivity is increased in the aggregates. Either the entire RP story should be taken out or it needs to be much better developed and tested.

I understand that the authors have established a protocol of generating Dicer1 conditional KO, however, just morphology remains the same without demonstrating more physiology

does not indicate this genetic manipulation will not affect muller cell or other cell type physiology and structures.

Under the title 'Mir9 is important for maintenance of the function of MG', I would imagine the authors will do mir9 knockdown in WT, or Dicer 1 MG KO mir9 rescue to see how it affects Bcan, or even exogenous Dicer injection to demonstrate how important mir9 is. But no related experiments were demonstrated.

Also, what about Bcan KO?, I feel Bcan KO is more direct way of linking Dicer1 function. Can RNAi and rescue experiment help explain the function? Does it affect Dicer or mir9, or mir9 precursor? I don't think the authors address anything related to this question either.

Therefore, it seems an empty summary.

The last section of the experiments attempted to explain the potential mechanism, but there is still no clear answer, and the disease model does not seem to connect with the discovery. Mir9 scramble is not shown anywhere in the paper either. Is there any effect if it were other species of mirRNAs?

Also, figure 4, analysis of Bcan is not showing significant changes. The attempt of linking mir9 and Bcan experiment was not convincing and is very important for this paper.

Reviewer #1:

1. The description of the mice is not cited or written correctly. This was a problem in a previous paper yet the authors make the same mistakes in this paper.

We received the Rlbp1-CreER mouse from the lab of Dr. Edward Levine, who first described the Rlbp1 promoter in the paper Vazques-Chona et al., 2009. However, we agree that this earlier report describes the construct used to generate the mice and is not about the mice per se. Therefore we edited the Methods accordingly.

We have used these mice in two previous studies and in both cases demonstrated MG-specific expression. (1) In Pollak et al, 2013 (Development), we demonstrated that this line shows MG-specific expression in a Supplemental Figure. (2) In Wohl and Reh, 2016 (SciRep) we characterized these mice with immunofluorescence and FACS purification using the following criteria: (1) FACS purified cells were plated and labeled for Muller glial markers; we found that all sorted and plated tdTomato+ cells from P11 mice were Sox9+, Sox2+ and Id1+, all markers for Muller glia. (2) To determine whether any neurons were labeled with the tdTomato reporter, we crossed the line to an Nrl-GFP reporter (to label rod photoreceptors, the most common retinal cell type) and did not find any GFP+tdTomato+ rods. (3) In immunofluorescence analysis of retinal sections from these mice, we found that the vast majority of labeled cells were Muller glia; no neurons were labeled, and the only other significant population of labeled cells were cells in the RPE and in the ciliary body, which we do not include in our dissections (cultures, explants).

However, as further documentation that this line of mice specifically targets the Muller glia, we now include additional images and data in this report (Extended Data Figure 1, a-e). We show that all tdTomato+ cells are only positive for glial markers such as GluSyn, Sox9, and Sox2. No PKC+ bipolar cells, Nrl+ photoreceptors or Sox2+ amacrine cells, were tdTomato+ (Figure 1 and Extended Data 1). In addition, we did not find any GFAP+ astrocytes positive for tdTomato. The new data was generated in collaboration with Dr. Ed Levine, whose lab generated the mice, and so he is now added as an author.

2. Why do only some MG respond with significant abnormal migration?

We do not know why Dicer1 deletion initially affects only a subset of the Muller glia. We speculate that both alleles of Dicer1 are not deleted in every Muller glia; in fact we now show with the RNA-Seq of FACS-purified glia that some Dicer1 remains in the cells (see new Extended Figure 4 g-h). Alternatively, it may be that Muller glia are not all alike (Roesch et al., 2008) and equally sensitive to the loss of Dicer1. Nevertheless, we only observed the patchy phenotype in retinas one month following Dicer1 deletion; six months after deletion, the vast majority of Muller glia are affected, leading to a severe phenotype of retinal disorganization. A paragraph discussing these possibilities is now in the Results section.

3. What happens if Dicer is deleted in adult MG?

Most of our analysis was concentrated in mice with Dicer1 deletion at two weeks, because our pilot studies showed a clear phenotype at this age; however, we have now done additional experiments where we deleted Dicer1 in older mice (2 months old). We observe a similar phenotype as that found in the mice with Dicer1 deletion at P11-14: an extended INL with Muller glial migration throughout the INL in some areas of the retina (see new Figure 3 and Extended Data Figure 2). Five months after Dicer deletion at P56-59, we found severe disruptions of the

retina with a clearly impaired outer limiting membrane and ONL thinning, in the central and peripheral retina. Nevertheless, the phenotype after 5 months was not that severe as it was in the mice with the Dicer1 deletion at P11-14. It is not clear why deletion of Dicer1 in older mice would have a less severe phenotype than when the gene is deleted in young mice, but we speculate that the accumulation of miRNAs in the older mice (Wohl and Reh, 2016, SciRep) takes longer to deplete. The new data is shown in Figure 3, and in the Extended Data section and described in Results.

4. How extensive is cell death upon loss of Dicer?

Dicer 1 deletion in Muller glia causes extensive cell loss after 6 and 12 months, especially of photoreceptors. We included a new Figure 2 and 3 showing the time course of photoreceptor loss and ONL thinning over time in the center and periphery of the retina. We did not see any obvious neuronal cell death one month after Dicer deletion. We also labeled the retinas for Caspase3+ to detect cells undergoing apoptosis, but found very few, suggesting that the loss of cells is either a very slow process, or it occurs rapidly sometime between 1-6 months after the deletion. The new Caspase3 labeling data is now shown in Figure 2.

Reviewer #2 (Remarks to the Author):

This paper presents some very interesting data on the role of miRNA in maintaining retinal Muller glia (MG) homeostasis and retinal architecture...Overall the manuscript is well written and provides significant new and interesting data that should be of interest to the Nature Communications readership, of interest not only to the retinal community but also to the more general glia/neuroscience communities. Despite the strengths of the manuscripts, below are some suggestions to clarify, add depth, and expand the presented work. Although clearly all the suggestions need not be taken, if the authors could address the ones that they feel most relevant to their work this could significantly increase the significance and interest of the work to the NC readership.

1. The work by Vázquez-Chona et al, which presumably describes the same promoter used by the authors, indicates that the Rlbp1 promoter is not totally MG-specific. This is probably not an issue for interpretation of the presented data, but should be acknowledged since it could theoretically contribute to the observed phenotype.

The reviewer is correct that Rlbp1 is also expressed in the RPE and the ciliary epithelium. Vazques-Chona et al reported weak GFP expression in the RPE, and we find that the Rlbp1-creER2:tdTomato mice show labeling in the at least some of the cells of the RPE and the ciliary epithelium. Therefore, it is possible then that Dicer1 is also deleted from the RPE and the ciliary epithelium, and that this contributes to the phenotype we observe in vivo. However, Ohana et al. have already characterized the effects of Dicer1 deletion from the RPE very early in development (E9.5 with onset of gene expression up to P11, not using an inducible cre), and the phenotype they observed did not include MG migration. However, we also observe PR death over time, which might be also due to Dicer1 deletion in the RPE cells. Since we observed the MG migration and aggregation in explants and cultures as well which were done in absence of the RPE, we are confident that our phenotype is predominantly because of the loss of Dicer1 in the MG. We compare and discuss the phenotypes in the Discussion. In addition, we do not find any other labeled cells in the retina in the Rlbp1-creER2:tdTomato line, and we added new images in Extended Data Figure 1 to better document this.

2. The authors state that in the induced Dicer-CKOMG retinas mice there is not only abnormal MG migration but also that the retinas show a 45% increase in INL thickness. However, the increase in tdTomato+ MG cells is only 23%, and MG make up only a fraction of the INL. This seems like an important finding and it would be helpful if the authors could describe in more detail the cellular basis of the INL thickening and at least discuss possible mechanisms.

We agree that the expansion of the retinal thickness is quite interesting; this is not confined to the INL, since we find that the ONL thickness is also increased (we document this in new Extended Data Figure 2h,i). We don't think this is due to an increase in cell number, but rather due to retinal stretching, possibly caused by changes in the Muller glia leading to changes in their tensile characteristics, (see MacDonald et al. for example). This occurs within the first month after Dicer deletion, and so is not due to the disorganization of the retina, but instead appears to be an early effect of the loss of Dicer1. We have added a paragraph to the Results/Discussion relevant to this phenomenon.

3. More exploration of the mechanisms of the expanded MG population. The authors mention increased proliferation as a likely possibility, but limited EdU experiments did not confirm this. More extensive exploration would be helpful. Since the phenotype is evident in explants, live imaging with time-lapse analysis could be very useful. Are results that they have observed different between explants and in vivo, as perhaps suggested by the text?

As suggested by the reviewer, we attempted time-lapse recordings in freshly isolated retinas over a period of 24 hours. During this period, we did not observe any mitotic divisions, consistent with the EdU labeling suggesting a very low rate of proliferation.

4. It is interesting that abnormalities in visual function, as measured by OptoMotry®, were evident as early as 2 weeks following miRNA depletion. Yet most of the molecular studies were performed at one month post Cre induction. Although of course all the molecular studies do not need to be repeated, since early time points could point more towards causative changes, as opposed to secondary changes, it would be helpful for the authors to test some of the miRNA and transcriptomic changes noted by their genome-wide assays at earlier time points. Looking at least some additional time points would be an important addition to the paper and should be done.

We focused on analyzing rather later time points than earlier time points, since we only begin to see changes in the retina at one month after the deletion. Since we did not observe a noticeable phenotype earlier than one month post-deletion, it is unlikely that we will see much change in glial gene expression at earlier time points. However, since full visual acuity is established around P25, it is possible that the Dicer deletion in the MG beginning P11-14 might have an impact on retinal maturation.

5. What was the actual level of Dicer1 expression in the purified induced MG cells? This seems important to confirm the KD, and this data should already be available from the RNA-Seq data, and would be good to also test by QPCR (sorry if this data is already presented and I missed it?)

We added additional data to Extended Data Figure 4 g,h which shows the overall Log2 CPM in

the wild type and the CKO (g). A sashimi plot in h shows the deleted exon 23, which is skipped in 19 out of 32 cases in the CKO, suggesting a 60% loss of exon 23.

6. Are all the primary datasets available in a publicly accessible database. This seems important, and presumably is a requirement of NC?

The primary data, including RNA-seq for FACS purified Muller glia, will be deposited to GEO and SRA upon acceptance of the manuscript. The datasets for generating each graph will also be available and hyperlinked to the Figure Legends.

7. It seems that some of the RNA-Seq data findings were confirmed by QPCR. Would be helpful to give more details. Were the samples tested by QPCR the same as the ones used for RNA-Seq, or were they independent samples – the latter would of course be better. How many replicates were tested? A scatter plot to compare fidelity between RNA-Seq and all QPCR data as a supplementary figure would be useful.

We sorted the MG from 40 retinas for the wt, 28 for the CKO and 26 for hets. We did lose the het sample for sending out for sequencing. We also did not have enough leftover material from the wt and used another wt sample, but the CKO sample was the same used for RNA-Seq. The biological replicates for the wt are 20 mice, and for the CKO 14 mice. The samples were pooled and used for Nanostrigs and RNA-Seq. The second wt sample for RT-qPCR was from 16 extra wt mice. We added a Scatter plot in Figure 4f showing a regression coefficient $R^2 = 0.6$. We also confirmed the downregulation of the highly expressed microRNAs via RT-qPCR (Figure 4b).

8. The data on glia aggregates and Brevican expression in retinitis pigmentosa retinas is potentially very interesting and important. But no details are provided – how many individual patient retinas were tested, what were the patient phenotypes, genotypes, how does the staining compare to control age-matched retinas. Although this work potentially adds significantly to the manuscript, if it can not be presented in more detail than it might be better not to present it at all.

We have added additional details as requested by the reviewer; however, we do not have extensive documentation on these samples, since they were collected several years ago by Dr. Ann Milam. We do not have age-matched control donor retinas, but we were able to show that the severity of the degeneration correlates well with the expression of Brevican in the Muller glia. We feel that this is a good control for showing Brevican upregulation in MG aggregates. Moreover, we found an increase in Glutamine Synthetase+ Brevican+ colocalized pixels per area analyzed between intact and disturbed regions in the sections and included this new data in Figure 7. We left the Figure in the main manuscript since we feel this adds potential relevance of our results to human disease (requested by another reviewer).

9. In the Discussion the authors state, “We found that miRNA depletion in MG leads to disorganization of the normal retinal architecture and ultimate impairment of visual acuity due to MG migration and aggregation.” Although the authors have clearly demonstrated retinal disorganization and MG migration and aggregation, I am not sure that they have experimentally shown that the disorganization and decreased visual acuity is mechanistically DUE TO the MG migration and aggregation.

We agree, that we don't know what is causing the decrease in visual acuity, and there is a small reduction in visual acuity even before we see anatomic disruption. Therefore we rephrased this sentence "(1) We found that miRNA depletion in MG leads to disorganization of the normal retinal architecture due to MG migration and aggregation. (2) We found excessive neuronal cell death 6 months after deletion in the ONL is accompanied by significant impairment of visual acuity.

Reviewer #3 (Remarks to the Author):

Overall, the authors show solid data linking miR9 to Bcan and the muller glia phenotype. The studies are well executed and the appropriate experimental controls are performed. The only limitation of the study is the relevance to disease or physiology. The reduction in vision is important but this is not surprising given the retinal disruption. A more direct connection to human retinal disease would strengthen the manuscript.

We thank the reviewer for this positive assessment. Although we cannot directly show that miRNAs are involved in the retinal disorganization that accompanies retinal degeneration, the phenotype we observe from Muller glial specific deletion of Dicer1 is quite similar to that observed in some cases of late staged Retinitis Pigmentosa. Moreover, we provide evidence of a correlation between Brevican expression and retinal disorganization, which had not previously been shown (Figure 7). There is at present not much known about the factors that drive the extensive retinal remodeling that occurs in late staged retinal degeneration, and we were led to Brevican by the mouse RNA-Seq data. So although additional studies will be necessary to better understand the relevance of Brevican to human retinal disease, our data point to a new research direction that could provide insight into this significant problem.

Reviewer #4 (Remarks to the Author):

(1) Unclear to me why the authors elect to delete Dicer with Tam injections at P11-P14. If the argument is that miRNAs are required for maintenance of mature Muller glia structure and function, I would like to see the effects of conditional deletion once animals reach adulthood. This also applies to the retinal explant culture experiments where the authors used P11 animals.

As noted above, most of our analysis was concentrated in mice with Dicer1 deletion at two weeks, because our pilot studies showed a clear phenotype at this age; however, we have now done additional experiments where we deleted Dicer1 in older mice (2 months old). We observe a similar phenotype as that found in the mice with Dicer1 deletion at P11-14: an extended INL with Muller glial migration throughout the INL in some areas of the retina (see new Figure 3e-l and Extended Data Figure 2j-m). Five months after Dicer deletion at P56-59, we found severe disruptions of the retina with a clearly impaired outer limiting membrane and ONL thinning, in the central and peripheral retina. Nevertheless, the phenotype after 5 months was not that severe as it was in the mice with the Dicer1 deletion at P11-14. It is not clear why deletion of Dicer1 in older mice would have a less severe phenotype than when the gene is deleted in young mice, but we speculate that the accumulation of miRNAs in the older mice (Wohl and Reh, 2016, SciRep) takes longer to deplete. The new data is shown and described in Results.

We used the explant model to match with the in vivo model. We also performed explant studies with adult tissue but adult tissue cannot be cultured successfully longer than 4 days and the phenotype takes longer to be that pronounced.

(2) Evaluation of overall retinal architecture by histology should be performed. Previous descriptive studies following Muller glia ablation have described rosette formation, which the aggregates by IHC seem to suggest.

We included new data about the overall retinal architecture showing the OLM is disrupted, photoreceptors die slowly leading to thinning of the ONL (see new Figure 2 and 3). We did see some rosette formation but not in every case of MG aggregation. However, the phenotype of the Dicer1 CKO retinas have many features in common with that described after MG ablation and we compare these in the Results and Discussion sections.

(3) The link between Muller glia and retinitis pigmentosa is glossed over and needs clarification. Are glia in control age-matched patient samples also immunoreactive for brevican?

As noted in the response to Reviewer #2, we have added additional details as requested by the reviewer; however, we do not have extensive documentation on these samples, since they were collected several years ago by Dr. Ann Milam. We do not have age-matched control donor retinas, but we were able to show that the severity of the degeneration correlates well with the expression of Brevican in the Muller glia. We feel that this is a good control for showing Brevican upregulation in MG aggregates. Moreover, we found an increase in Glutamine Synthetase+ Brevican+ colocalized pixels per area analyzed between intact and disturbed regions in the sections and included this new data in Figure 7.

(4) I understand that the authors have established a protocol of generating Dicer1 conditional KO, however, just morphology remains the same without demonstrating more physiology does not indicate this genetic manipulation will not affect muller cell or other cell type physiology and structures.

We are not sure what the reviewer is getting at here. The deletion of Dicer1 in this study is specific to the Muller glia. The morphology of the Muller glia is clearly abnormal and we document changes in RNA and miRNA expression levels. Although we did not carry out electrophysiology on the cells, that was not the focus of our investigation. We know of no study that links changes in Muller glial electrophysiology to changes in their migratory behavior.

(5) Under the title 'Mir9 is important for maintenance of the function of MG', I would imagine the authors will do mir9 knockdown in WT, or Dicer 1 MG KO mir9 rescue to see how it affects Bcan, or even exogenous Dicer injection to demonstrate how important mir9 is. But no related experiments were demonstrated.

The reviewer is not correct on this point; the subheading in the previous version was “mir-9 is important for maintenance of normal MG.” Our focus was on the morphology of the cells and not on their function. Moreover, as suggested by the reviewer, we described a Dicer1 CKO rescue experiment with miR-9 in this section, shown in Figure 6. However, we agree with the reviewer that the original subheading might not be best characterize the data in this section and so we have changed it to “miR-9 targets Brevican”.

(6) Also, what about Bcan KO?, I feel Bcan KO is more direct way of linking Dicer1 function. Can RNAi and rescue experiment help explain the function? Does it affect Dicer or mir9, or mir9 precursor?

It is not clear what we would learn from the Bcan knockout. Bcan is not normally expressed in Muller glia in the wild type retina, and so we would only predict a result from this experiment if the Bcan knockout was carried out on the same mice as the conditional deletion of Dicer1. Generating mice with homozygous conditional deletions in these two genes would require substantial effort and we feel is beyond the scope of the current manuscript. However, as suggested by the reviewer, we link Bcan with miR-9 and Dicer1 deletion with the following experiments: (1) Over-expression of Bcan with AAV leads to a similar phenotype as MG specific deletion of Dicer1; (2) blocking Bcan leads to a rescue of the phenotype; and (3) treating the Dicer1 CKO MG with miR-9 mimics also rescues the phenotype.

The last section of the experiments attempted to explain the potential mechanism, but there is still no clear answer, and the disease model does not seem to connect with the discovery. Mir9 scramble is not shown anywhere in the paper either. Is there any effect if it were other species of mirRNAs?

We did use control mimics from *C.elegans* (see Supplemental Table 1), which is essentially like a scrambled sequence, and a control used by many studies. We did use control mimics for every experiment, but in the previous version we were not explicit in each experiment; we thank the reviewer for this correction, and we now define the control mimic for each experiment. Also we did use other miRs, miR-181a, miR-125b-5p, and let7a, and found that these did not have the same effects as miR-9, thus providing additional control for the specificity of the effects. The data reporting about other miRs is shown in Figure 6 and Extended Data Figure 6.

Also, figure 4, analysis of Bcan is not showing significant changes. The attempt of linking mir9 and Bcan experiment was not convincing and is very important for this paper.

The reviewer was correct that this link was not as well established as some of the other parts of our study. We have now provided several lines of evidence to link miR-9 and Bcan:

1. The 3' UTR of Bcan has a miR-9 consensus binding site (Figure 6a)
2. A Bcan 3'UTR sensor is significantly reduced by miR-9 mimics (Figure 6b-e)
3. miR-9 mimics rescue the main features of the Dicer-CKO Muller glial phenotype: the formation of aggregates (Figure 6f-j) and the migration to ectopic retinal laminae (Figure 6k-o). Additional data relevant to all these points as well as additional controls are shown in Extended data Figures 5 and 6.

REVIEWERS' COMMENTS:

Reviewer #3 (Remarks to the Author):

the authors have addressed the comments.

Reviewer #4 (Remarks to the Author):

the revisions are sufficient for publications